# WDR90 is a centriolar microtubule wall protein important for centriole architecture integrity

Emmanuelle Steib[1], Marine H Laporte[1], Davide Gambarotto[1], Natacha Olieric[2], Celine Zheng[2†], Susanne Borgers[1], Vincent Olieric[3], Maeva Le Guennec[1], France Koll[4], Anne-Marie Tassin[4], Michel O Steinmetz[2,5], Paul Guichard[1]*, Virginie Hamel[1]*

[1]University of Geneva, Department of Cell Biology, Sciences III, Geneva, Switzerland; [2]Laboratory of Biomolecular Research, Division of Biology and Chemistry, Paul Scherrer Institut, Villigen, Switzerland; [3]Swiss Light Source, Paul Scherrer Institut, Villigen, Switzerland; [4]Institute for Integrative Biology of the Cell (I2BC), CEA, CNRS, Univ. Paris Sud, Université Paris-Saclay, Gif sur Yvette, France; [5]Biozentrum, University of Basel, Basel, Switzerland

**Abstract** Centrioles are characterized by a nine-fold arrangement of microtubule triplets held together by an inner protein scaffold. These structurally robust organelles experience strenuous cellular processes such as cell division or ciliary beating while performing their function. However, the molecular mechanisms underlying the stability of microtubule triplets, as well as centriole architectural integrity remain poorly understood. Here, using ultrastructure expansion microscopy for nanoscale protein mapping, we reveal that POC16 and its human homolog WDR90 are components of the microtubule wall along the central core region of the centriole. We further found that WDR90 is an evolutionary microtubule associated protein. Finally, we demonstrate that WDR90 depletion impairs the localization of inner scaffold components, leading to centriole structural abnormalities in human cells. Altogether, this work highlights that WDR90 is an evolutionary conserved molecular player participating in centriole architecture integrity.

*For correspondence:
paul.guichard@unige.ch (PG);
virginie.hamel@unige.ch (VH)

Present address: †Department of Biochemistry, University of Oxford, Oxford, United Kingdom

Competing interests: The authors declare that no competing interests exist.

## Introduction

Centrioles and basal bodies (referred to as centrioles from here onwards for simplicity) are conserved organelles important for the formation of the centrosome as well as for templating cilia and flagella assembly (*Bornens, 2012*; *Breslow and Holland, 2019*; *Conduit et al., 2015*; *Ishikawa and Marshall, 2011*). Consequently, defects in centriole assembly, size, structure and number lead to abnormal mitosis or defective ciliogenesis and have been associated with several human pathologies such as ciliopathies and cancer (*Gönczy, 2015*; *Nigg and Holland, 2018*; *Nigg and Raff, 2009*). For instance, centriole amplification, a hallmark of cancer cells, can result from centriole fragmentation in defective, over-elongated centrioles (*Marteil et al., 2018*).

Centrioles are characterized by a nine-fold radial arrangement of microtubule triplets, are polarized along their long axis, and can be divided in three distinct regions termed proximal end, central core and distal tip (*Hamel et al., 2017*). Each region displays specific structural features such as the cartwheel on the proximal end, which is crucial for centriole assembly (*Nakazawa et al., 2007*; *Strnad et al., 2007*) or the distal appendages at the very distal region, essential for membrane docking during ciliogenesis (*Tanos et al., 2013*). The central core region of the centriole is defined by the presence of a circular inner scaffold thought to maintain the integrity of microtubule triplets under compressive forces (*Le Guennec et al., 2020*). Using cryo-tomography, we recently showed that the

**eLife digest** Cells are made up of compartments called organelles that perform specific roles. A cylindrical organelle called the centriole is important for a number of cellular processes, ranging from cell division to movement and signaling. Each centriole contains nine blades made up of protein filaments called microtubules, which link together to form a cylinder. This well-known structure can be found in a variety of different species. Yet, it is unclear how centrioles are able to maintain this stable architecture whilst carrying out their various different cell roles.

In early 2020, a group of researchers discovered a scaffold protein at the center of centrioles that helps keep the microtubule blades stable. Further investigation suggested that another protein called WDR90 may also help centrioles sustain their cylindrical shape. However, the exact role of this protein was poorly understood.

To determine the role of WDR90, Steib et al. – including many of the researchers involved in the 2020 study – used a method called Ultrastructure Expansion Microscopy to precisely locate the WDR90 protein in centrioles. This revealed that WDR90 is located on the microtubule wall of centrioles in green algae and human cells grown in the lab. Further experiments showed that the protein binds directly to microtubules and that removing WDR90 from human cells causes centrioles to lose their scaffold proteins and develop structural defects.

This investigation provides fundamental insights into the structure and stability of centrioles. It shows that single proteins are key components in supporting the structural integrity of organelles and shaping their overall architecture. Furthermore, these findings demonstrate how ultrastructure expansion microscopy can be used to determine the role of individual proteins within a complex structure.

inner centriole scaffold forms an extended helix covering ~70% of the centriole length and that is rooted at the inner junction between the A and B microtubules (*Figure 1A,B*). This connection consists of a stem attaching the neighboring A and B microtubules and three arms extending from the same stem toward the centriolar lumen (*Le Guennec et al., 2020*; *Figure 1A,B*). The stem of the inner scaffold has been detected in *Paramecium tetraurelia*, *Chlamydomonas reinhardtii* and human centrioles, suggesting that it represents an evolutionary conserved structural feature.

The molecular identity of some components of the inner scaffold has been uncovered using Ultrastructure Expansion Microscopy (U-ExM), which allows nanoscale localization of proteins within structural elements (*Gambarotto et al., 2019*). Notably, the centriolar proteins POC1B, FAM161A, POC5 and Centrin have been shown to localize to the inner scaffold along the microtubule blades in human cells (*Le Guennec et al., 2020*). Moreover, these proteins form a complex that can bind to microtubules through the microtubule-binding protein FAM161A (*Le Guennec et al., 2020*; *Zach et al., 2012*). Importantly, a subset of these proteins has been shown to be important, such as POC5 for centriole elongation (*Azimzadeh et al., 2009*) as well as POC1B for centriole and basal body integrity (*Pearson et al., 2009*; *Venoux et al., 2013*). This observation highlights the role of the inner scaffold structure in providing stability to the entire centriolar microtubule wall organization. However, the exact contribution of the inner scaffold to microtubule triplets stability and how the inner scaffold is connected to the microtubule blade is unknown.

We recently identified the conserved proteins POC16/WDR90 as proteins localizing to the central core region in both *Chlamydomonas reinhardtii* and human centrioles (*Hamel et al., 2017*). Impairing POC16 or WDR90 functions has been found to affect ciliogenesis, suggesting that POC16/WDR90 may stabilize the microtubule wall, thereby ensuring proper flagellum or cilium assembly (*Hamel et al., 2017*). Interestingly, POC16 has been proposed to be at the inner junction between the A and B microtubules (*Yanagisawa et al., 2014*) through its sequence identity with FAP20, an axonemal microtubule doublet inner junction protein of *Chlamydomonas reinhardtii* flagella (*Dymek et al., 2019*; *Ma et al., 2019*; *Owa et al., 2019*; *Yanagisawa et al., 2014*). As the stem connects the A- and B-microtubules interface, these observations suggest that POC16/WDR90 may connect the inner scaffold to the microtubule triplet through this stem structure (*Figure 1C*), thus ensuring integrity of the centriole architecture.

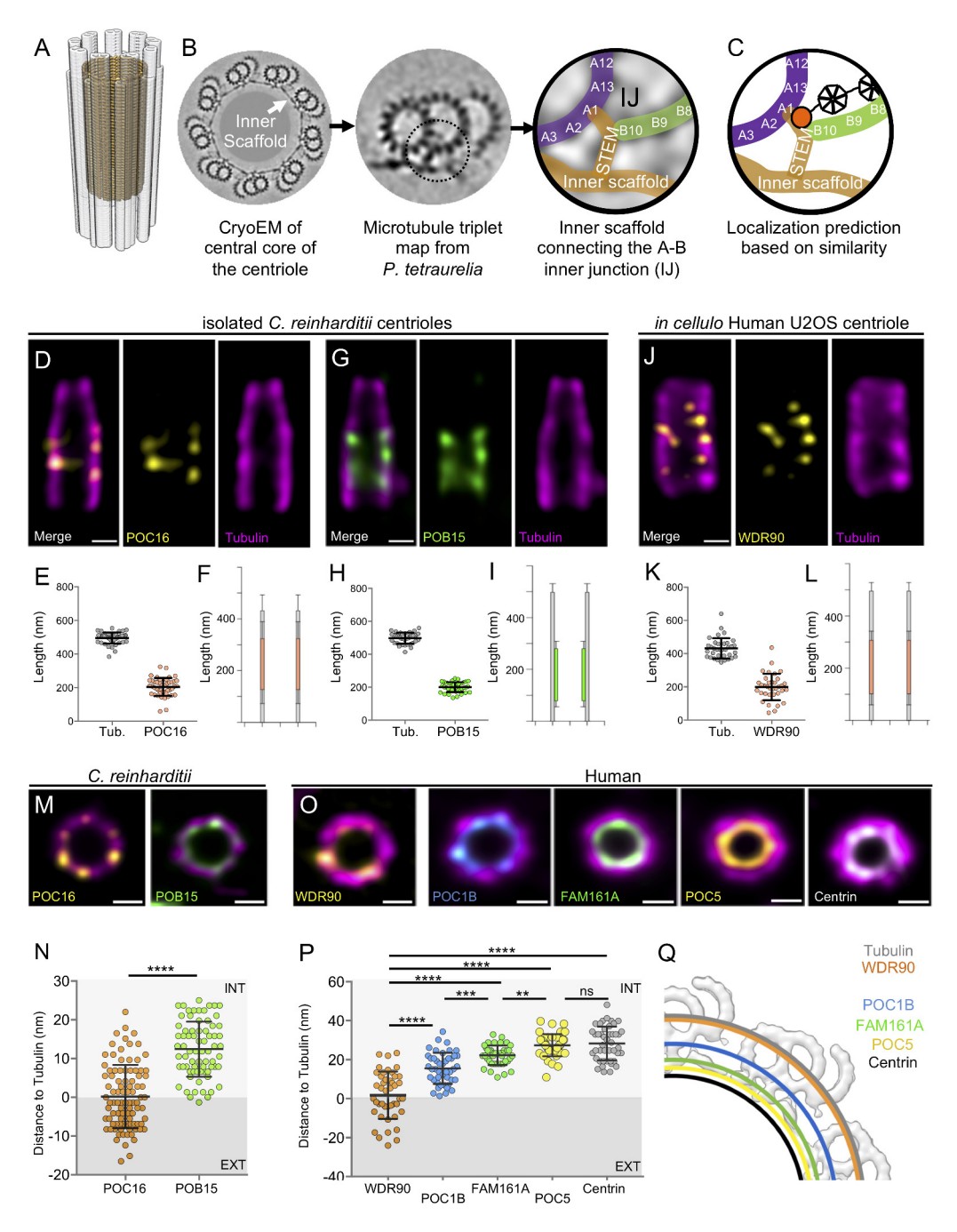

**Figure 1.** POC16/WDR90 is a conserved central core microtubule wall component. (**A**) 3D representation of a centriole highlighting the centriolar microtubule wall in light grey and the inner scaffold in yellow. (**B**) Cryo-EM image of the central core of *Paramecium tetraurelia* centrioles from which a microtubule triplet map has been generated (*Le Guennec et al., 2020*). Schematic representation of the inner junction (IJ) between A- and B-microtubules connecting the inner scaffold. (**C**) Schematic localization of POC16/WDR90 proteins within the IJ based on its similarity to FAP20. Purple: A-microtubule, green: B microtubule, yellow/gold: inner scaffold and stem, orange: DUF667 domain positioned at the IJ. (**D**) Isolated U-ExM expanded *Chlamydomonas* centriole stained for POC16 (yellow) and tubulin (magenta), lateral view. Scale bar: 100 nm. (**E**) Respective lengths of tubulin and POC16 based on D. Average +/- SD: Tubulin: 495 nm +/- 33, POC16: 204 nm +/- 53, n = 46 centrioles from three independent experiments. (**F**) POC16 length coverage and positioning: 41% +/- 11, n = 46 centrioles from three independent experiments. (**G**) Expanded isolated *Chlamydomonas* centriole stained for POB15 (green) and tubulin (magenta), lateral view. Scale bar: 100 nm. (**H**) Respective length of tubulin and POB15 based on G. Average +/- SD: tubulin = 497 nm +/- 33, POB15 = 200 nm +/- 30, n = 39 centrioles from three independent experiments. (**I**) POB15 length coverage and positioning: 40% +/- 6, n = 39 centrioles from three independent experiments. (**J**) Expanded human U2OS centriole stained for WDR90 (yellow) and

*Figure 1 continued on next page*

*Figure 1 continued*

tubulin (magenta), lateral views. (**K**) Respective lengths of tubulin and WDR90 based on J. Average +/- SD: Tubulin: 432 nm +/- 62, WDR90: 200 nm +/- 80, n = 35 from three independent experiments. (**L**) WDR90 length coverage and positioning: 46% +/- 17, n = 35 from three independent experiments. (**M**) Isolated U-ExM expanded *Chlamydomonas* centriole stained for tubulin (magenta) and POC16 (yellow) or POB15 (green), top views. Scale bar: 100 nm. (**N**) Distance between the maximal intensity of tubulin and the maximal intensity of POC16 (orange) or POB15 (green) based on M. Average +/- SD: POC16 = 0 nm +/- 8, POB15 = 12 nm +/- 7. n > 75 measurements/condition from 30 centrioles from three independent experiments. EXT: exterior or the centriole, INT: interior. Mann-Whitney test ****p<0.0001. (**O**) Expanded U2OS centriole stained for WDR90 (yellow) and tubulin (magenta), or for core proteins POC1B (blue), FAM161A (green), POC5 (yellow) or Centrin (white). Data set from *Le Guennec et al., 2020*, top views, Scale bars: 100 nm. (**P**) Distance between the maximal intensity of tubulin and the maximal intensity of WDR90 (orange) or POC1B (blue), FAM161A (green), POC5 (yellow) or Centrin (grey) based on O. Average +/- SD: WDR90 = 2 nm +/- 12, POC1B = 15 nm+/-8, FAM161A = 22 nm+/-5, POC5 = 27 nm +/- 6 and Centrin = 28 nm+/-9. n = 45 measurements/condition from 15 to 30 centrioles per condition from three independent experiments. One-way ANOVA and Holm-Sidak's multiple comparisons ns p>0.05, **p<0.01, ***p<0.001, ****p<0.0001. (**Q**) Position of WDR90 relative to the four inner scaffold components placed on the cryo-EM map of the *Paramecium* central core region (top view) (adapted from *Le Guennec et al., 2020*).

In this study, using a combination of cell biology, biochemistry and Ultrastructure Expansion Microscopy (U-ExM) approaches, we establish that the conserved POC16/WDR90 proteins localize on the centriolar microtubule wall in the central core region of both *Chlamydomonas* and human cells. We further demonstrate that WDR90 is a microtubule-binding protein and that loss of this protein impairs the localization of inner scaffold components and leads to slight centriole elongation, impairment of the canonical circular shape of centrioles as well as defects in centriolar architecture integrity.

## Results

### POC16/WDR90 is a conserved microtubule wall component of the central core region

To test the hypothesis that POC16/WDR90 is a microtubule triplet component, we analyzed its distribution using U-ExM that allows nanoscale mapping of proteins inside the centriole (*Gambarotto et al., 2019*; *Le Guennec et al., 2020*). We observed first in *Chlamydomonas reinhardtii* isolated centrioles that the endogenous POC16 longitudinal fluorescence signal is restricted to the central core region as compared to the tubulin signal, which depicts total centriolar length (*Figure 1D–F*). From top viewed centrioles, we measured the distance between both POC16 and tubulin maximal intensity signal from the exterior to the interior of the centriole and found that POC16 localizes precisely on the microtubule wall in the central core region of *Chlamydomonas* centrioles (*Figure 1M,N*, average distance between POC16 and tubulin Δ = 0 nm +/- 8). As a control, we could recapitulate the internal localization along the microtubule wall of POB15, another central core protein (*Figure 1G–I* and *Figure 1M,N*, average distance between POB15 and tubulin Δ = 12 nm +/- 7) as previously reported using immunogold-labeling (*Hamel et al., 2017*). In human centrioles, the POC16 human homolog WDR90 localizes similarly to POC16 on the centriolar microtubule wall, demonstrating the evolutionary conserved restricted localization of POC16/WDR90 on microtubule triplets in the central core region of centrioles (*Figure 1J–L*). Of note, POC16 and WDR90 display a punctate distribution that we hypothesize to be due to the poor quality of the antibody.

Next, we compared the relative position of WDR90 from top view centrioles to previously described inner scaffold components (*Figure 1O–Q*) (see Materials and methods). We found that while WDR90 precisely localizes to the centriolar microtubule wall (*Figure 1P*, average distance between WDR90 and tubulin: Δ = 2 nm +/12), POC1B, FAM161A, POC5 and Centrin signals were shifted toward the centriole lumen in comparison to the tubulin signal, as previously reported (*Figure 1P*, Δ = 15 nm +/- 8; 22 nm +/- 5; 27 nm +/- 6 and 28 nm +/- 9, respectively) (*Le Guennec et al., 2020*). These results demonstrate that WDR90 longitudinal distribution is similar to the inner scaffold components but its localization on the microtubule wall suggests that WDR90 is a component of the centriolar microtubule triplet of the central core region.

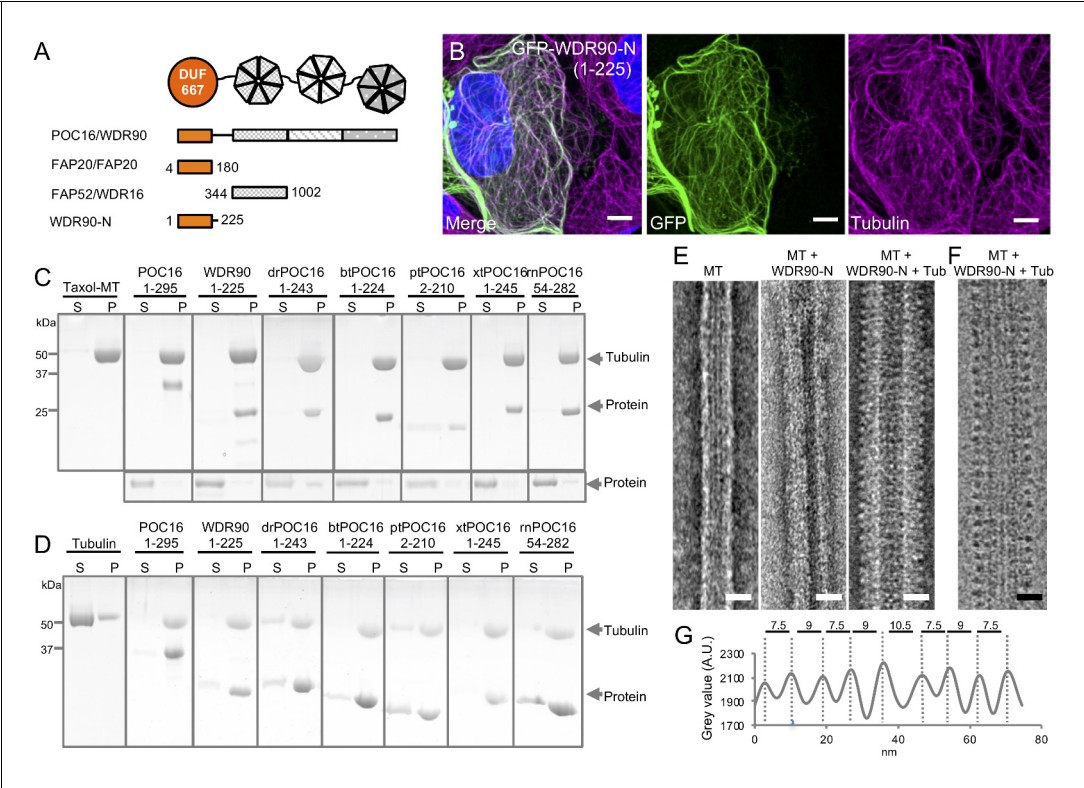

**Figure 2.** WDR90/POC16-DUF667 directly binds both microtubules and tubulin. (see also *Figure 2—figure supplements 1–3*). (**A**) Schematic of WDR90/POC16 conservation domains with the *Chlamydomonas* cilia proteins FAP20 and FAP52/WDR16. DUF667 domain is in orange and WD40 repeats are in grey. (**B**) Human U2OS cells transiently overexpressing GFP-WDR90-N (1-225) stained for GFP (green) and tubulin (magenta). Scale bar: 5 μm. (**C, D**) Coomassie-stained SDS-PAGE of pelleting assays performed in vitro with taxol-stabilized microtubules (**C**), and free tubulin (**D**), in the presence of different recombinant POC16/WDR90-DUF667 protein orthologs (related to *Figure 2—figure supplement 1A, B*). The solubility of proteins alone was assessed in parallel to the microtubule-pelleting assay. All tested proteins were soluble under the tested condition (bottom panel). (**E**) Electron micrographs of negatively stained taxol-stabilized microtubules alone (MT) or subsequently incubated with recombinant WDR90-N (1-225) alone (MT + WDR90-N) or in combination with tubulin (MT + WDR90-N + Tub). Scale bar: 25 nm (**F**) Cryo-electron micrograph of taxol-stabilized microtubules subsequently incubated with recombinant WDR90-N (1-225) and tubulin (MT + WDR90-N + Tub). Scale bar: 25 nm (**G**) Periodicity of complexed WDR90-N (1-225)-tubulin oligomers bound to the microtubule shown in (**F**).

The online version of this article includes the following figure supplement(s) for figure 2:

**Figure supplement 1.** POC16 conservation across species.
**Figure supplement 2.** Model prediction of POC16 Nter.
**Figure supplement 3.** POC16 and WDR90 bind microtubules.

## POC16/WDR90 is an evolutionary conserved microtubule-associated protein

Proteins of the POC16/WDR90 family consist of an N-terminal DUF667-containing domain (domain of unknown function), similar to the ciliary inner junction protein FAP20 (*Figure 2—figure supplement 1A*; *Yanagisawa et al., 2014*), followed by multiple WD40 repeats that form β-propeller structures (*Figure 2A* and *Figure 2—figure supplement 1B*; *Xu and Min, 2011*).

First, we wanted to probe the evolutionary conservation of POC16/WDR90 family members as centriolar proteins. To this end, we raised an antibody against *Paramecium tetraurelia* POC16 and confirmed its localization at centrioles similarly to what we found in *Chlamydomonas reinhardtii* and human cells (*Figure 2—figure supplement 1C*; *Hamel et al., 2017*).

Further driven by its predicted similarity to the microtubule associated protein FAP20 (*Khalifa et al., 2020*) and the underlying hypothesis that POC16/WDR90 proteins might be joining

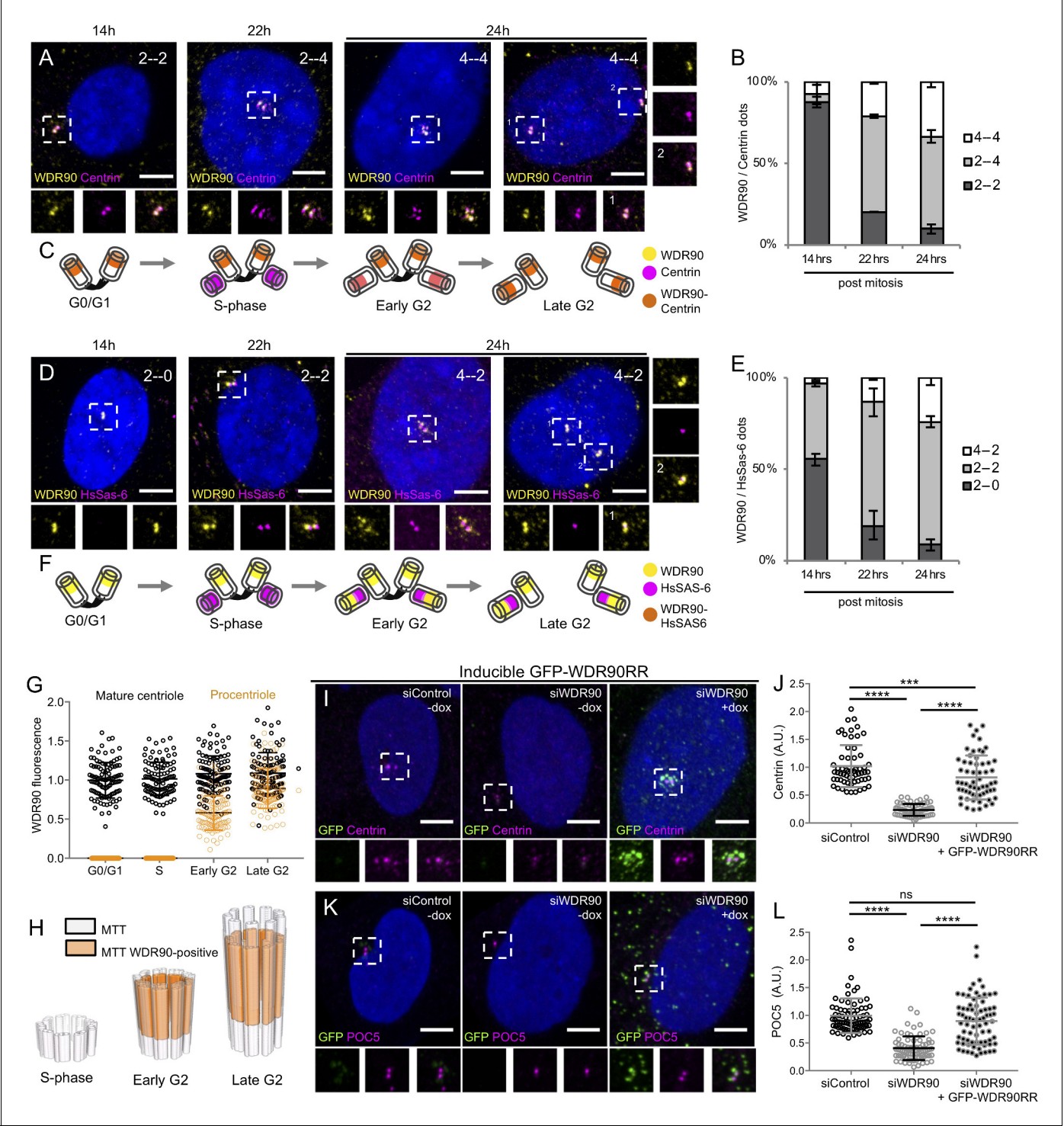

**Figure 3.** WDR90 is recruited in G2 and is important for Centrin and POC5 recruitment to centrioles (See also *Figure 3—figure supplements 1* and *2*). (A) Human RPE1 p53- cells synchronized by mitotic shake-off, fixed at different time points for different cell-cycle stages (related to *Figure 3—figure supplement 1A, B*) and stained with WDR90 (yellow) and Centrin (magenta). DNA is in blue. Dotted white squares correspond to insets. Numbers on the top right indicate respectively WDR90 and Centrin numbers of dots. Scale bar: 5 μm. (B) Percentage of cells with the following numbers of WDR90/Centrin dots based on A, n = 300 cells/condition from three independent experiments. Average +/- SD: refer to *Figure 3—source data 1*. (C) Model for WDR90 and Centrin incorporation during centriole biogenesis based on A. (D) Human RPE1 p53- cells synchronized by mitotic shake-off, fixed at different time points for different cell-cycle stages and stained with WDR90 and HsSAS-6. Scale bar: 5 μm. (E) Percentage of cells with the following numbers of WDR90 and HsSAS-6 based on D, n = 300 cells/condition from three independent experiments. Average +/- SD: refer to *Figure 3—source*

*Figure 3 continued on next page*

*Figure 3 continued*

*data 2*. (F) Model for WDR90 and HsSAS-6 incorporation during centriole biogenesis based on D. (G) WDR90 fluorescence intensity at centrioles according to cell cycle progression, n = 45 cells/condition from three independent experiments. Black circle represents WDR90 at mature centrioles, orange circle represents WDR90 at procentrioles. (H) 3D Schematic representation of WDR90 incorporation during centriole biogenesis according to cell cycle progression based on G. (I, K) Human U2OS GFP-WDR90 RNAi-resistant version (GFP-WDR90RR) inducible stable cell line treated with control or *wdr90* siRNA and stained for either GFP and Centrin (I) or GFP and POC5 (K) Dotted white squares indicate insets. - and + dox indicates induction of GFP-WDR90RR expression. Scale bar: 5 µm. (J) Centrosomal Centrin fluorescence intensity based on I, n = 60 cells/condition from three independent experiments. Average +/- SD (A.U.): Control – dox = 1.02 +/- 0.4, siWDR90 – dox = 0.23+/- 0.1, siWDR90 + dox = 0.82 +/- 0.4. Statistical significance assessed by one-way ANOVA and Holm-Sidak's multiple comparisons (\*\*\*p<0.001, \*\*\*\*p<0.0001). (L) Centrosomal POC5 fluorescence intensity based on K, n = 75 cells/condition from three independent experiments. Average +/- SD (A.U.): Control – dox = 0.99 +/- 0.3, siWDR90 – dox = 0.41+/- 0.2, siWDR90 + dox = 0.89 +/- 0.5. One-way ANOVA and Holm-Sidak's multiple comparisons (ns p>0.05, \*\*\*\*p<0.0001).

The online version of this article includes the following source data and figure supplement(s) for figure 3:

**Source data 1.** Percentage of cells with the following number of dots/cell respectively for WDR90 and Centrin.

**Source data 2.** Percentage of cells with the following number of dots/cell respectively for WDR90 and HsSAS-6.

**Figure supplement 1.** WDR90 is a satellite and centriolar protein.

**Figure supplement 1—source data 1.** Percentage of cells in each phase of the cell cycle according to post-mitotic time point.

**Figure supplement 2.** Depletion of WDR90 impairs Centrin and POC5 localization at centrioles.

**Figure supplement 2—source data 1.** Percentage of cells displaying 0, 1, 2 or 4 dots of WDR90 based on the number of Centrin dots in U2OS cells treated with control or *wdr90* siRNA.

**Figure supplement 2—source data 2.** Percentage of cells displaying 0, 1, 2 or 4 dots of POC5 based on the number of HsSas-6 dots in U2OS cells treated with control or *wdr90* siRNA.

A and B microtubules as well as by their precise localization on the microtubule wall (*Figure 1*), we first set out to understand the structural identity between the predicted structures of POC16-DUF667 domain to the recently published near atomic structure of FAP20 from flagella microtubule doublets (*Khalifa et al., 2020*; *Ma et al., 2019*; *Figure 2—figure supplement 2A–C*). Strikingly, we observed high similarities between the two structures, suggesting similar biological functions at the inner junction. Moreover, we fitted POC16 model prediction into FAP20 cryo-EM density map and found a good concordance, further hinting for a conserved localization at the level of the microtubule triplet (*Figure 2—figure supplement 2D*).

Prompted by this result, we then tested whether POC16/WDR90 proteins, similar to FAP20, can bind microtubules both in human cells as well as in vitro. To do so, we overexpressed the N-terminal part of WDR90 and POC16 comprising the DUF667 domain (WDR90-N(1-225) and POC16(1-295), respectively) fused to GFP in U2OS cells and found that this region is sufficient to decorate cytoplasmic microtubules (*Figure 2B* and *Figure 2—figure supplement 3A*). We next tested whether overexpressing such a WDR90-N-terminal fragment could stabilize microtubules. To this end, we analyzed the microtubule network in cells overexpressing mCherry-WDR90-N after depolymerizing microtubules through a cold shock treatment (*Figure 2—figure supplement 3B–D*). We found that while low expressing cells did not maintain a microtubule network, high expressing cells did. This suggests that WDR90-N can stabilize microtubules. In contrast, we observed that full-length WDR90 fused to GFP only anecdotally binds microtubules. This observation suggests a possible autoinhibition conformation of the full-length protein and/or to interacting partners preventing microtubule binding in the cytoplasm (*Figure 2—figure supplement 3E*).

Next, we determined whether different POC16/WDR90 N-terminal domains directly bind to microtubules in vitro and whether this function has been conserved in evolution. Bacterially expressed, recombinant POC16/WDR90 DUF667 domains from seven different species were purified and their microtubule interaction ability was assessed using a standard microtubule-pelleting assay (*Figure 2—figure supplement 1A* and *Figure 2C*). We found that every POC16/WDR90 DUF667 domain directly binds to microtubules in vitro. This interaction was further confirmed using negative staining electron microscopy, where we could observe recombinant WDR90-N localizing on in vitro polymerized microtubules (*Figure 2E*).

We next investigated whether POC16/WDR90 DUF667 domain could also interact with free tubulin dimers, considering that closure of the inner junction between the A and B microtubules

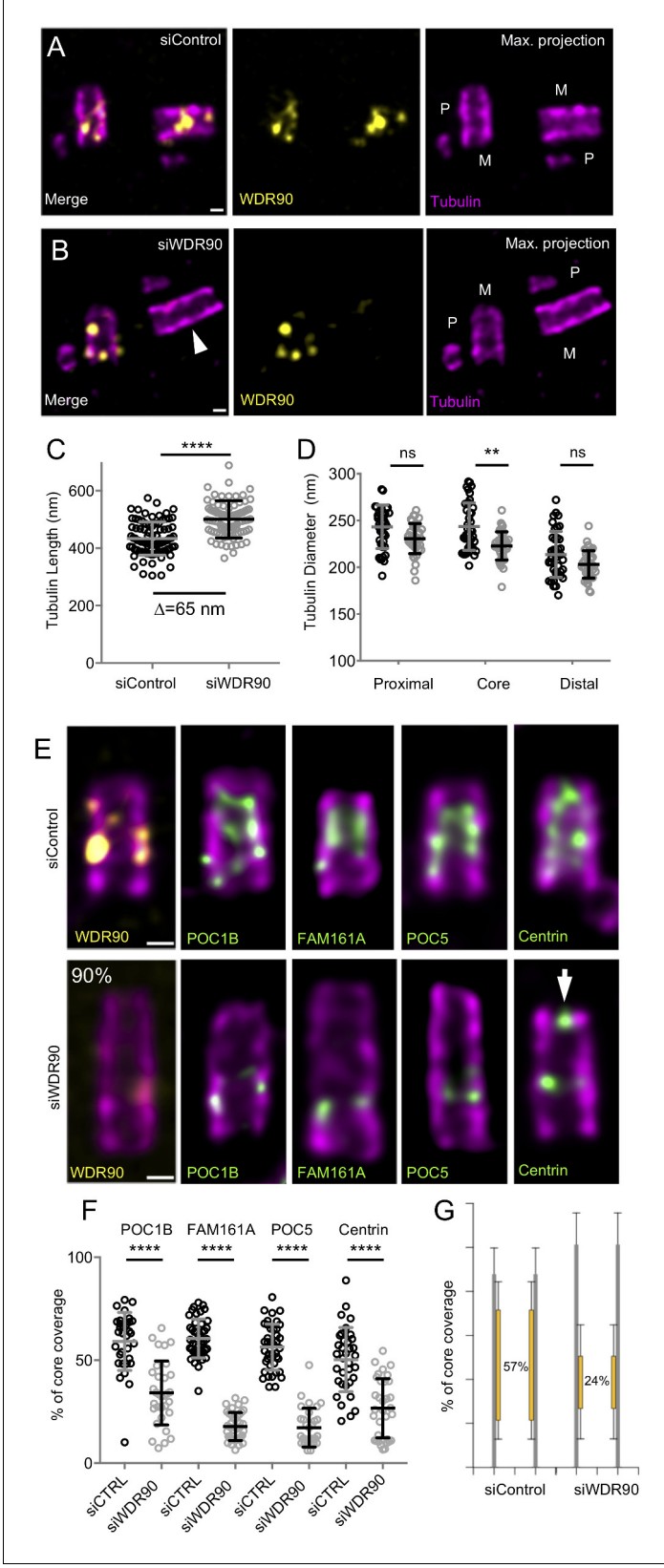

**Figure 4.** WDR90 is crucial for inner scaffold components localization (see also *Figure 4—figure supplement 1*). (**A, B**) Expanded centrioles from S-phase U2OS cells treated with either control (**A**) or *wdr90* siRNA (**B**) stained for tubulin (magenta) and WDR90 (yellow). M stands for mature centriole and P for procentriole. White arrowhead
*Figure 4 continued on next page*

*Figure 4 continued*

points to WDR90-depleted centriole. Scale bar: 100 nm. (**C**) Tubulin length in nm, n = 90 centrioles/condition from three independent experiments. Average +/- SD: siControl = 434 nm +/- 58, siWDR90 = 500 nm +/- 65. Mann-Whitney p<0.0001. Note that only efficiently depleted centrioles were counted. (**D**) Tubulin diameter measured in the proximal, central core and distal regions of expanded centrioles in control (black circles) and *wdr90* siRNA (siWDR90, grey circles). n = 42 and 43 centrioles for siControl and siWDR90 from two independent experiments, respectively. Averages +/- SD: refer to *Figure 4—source data 1*. One-way ANOVA and Holm-Sidak's multiple comparisons (ns p<0.05, **p<0.01). (**E**) Expanded U2OS centrioles treated with either control or *wdr90* siRNA stained for tubulin (magenta) and WDR90 (yellow) or POC1B, FAM161A, POC5 or Centrin (inner scaffold components: green). White arrow indicates the distal localization of Centrin. Scale bar: 100 nm. (**F**) Inner scaffold protein length expressed as a percentage of the total tubulin length, n > 30 centrioles/condition from three independent experiments. Average +/- SD: refer to *Figure 4—source data 2*. One-way ANOVA and Holm-Sidak's multiple comparisons (****p<0.0001). (**G**) Average core length coverage. Average +/- SD: siControl = 57% +/- 13; siWDR90 = 24% +/- 14.

The online version of this article includes the following source data and figure supplement(s) for figure 4:

**Source data 1.** Diameter at proximal, core and distal region of the centriole.
**Source data 2.** Inner scaffold proteins coverage.
**Figure supplement 1.** WDR90 depletion affects mainly inner scaffold components.

necessitates two microtubule/tubulin-binding sites as recently reported for FAP20 (*Ma et al., 2019*). We observed that all POC16/WDR90 DUF667 orthologs directly interact with tubulin dimers, generating oligomers that pellet under centrifugation (*Figure 2D*). We then tested whether the DUF667 domain could still interact with tubulin once bound to microtubules. We subsequently incubated either WDR90-N or POC16(1-295) pre-complexed with microtubules with an excess of free tubulin and analyzed their structural organization by electron microscopy (*Figure 2E,F* and *Figure 2—figure supplement 3F,G*). We observed an additional level of decoration due to the simultaneous binding of the DUF667 domains with tubulin and microtubules (*Figure 2E,F* and *Figure 2—figure supplement 3F,G*). Furthermore, we revealed a 8.5 nm periodical organization of tubulin-WDR90-N oligomers on microtubules (*Figure 2G*), similar to the recent high-resolution structure of the ciliary microtubule doublet showing that monomeric FAP20 interacts with both A- and B-microtubules every 8 nm at the inner junction (*Khalifa et al., 2020*; *Ma et al., 2019*). Due to its similarity, it is tempting to speculate that the DUF667 domain of POC16/WDR90 is also monomeric, however it is also possible that WDR90 forms a homodimer capable of interacting with the microtubules and tubulin.

Based on these results, we concluded that POC16/WDR90 is an evolutionary conserved microtubule/tubulin-interacting protein with the capacity to connect microtubules, a functional prerequisite for an inner junction protein that simultaneously interacts with the A and B microtubules.

## WDR90 is recruited in G2 during centriole core elongation

We next assessed whether WDR90 recruitment at centrioles is correlated with the appearance of inner scaffold proteins during centriole biogenesis. In cycling human cells, centrioles duplicate only once per cell cycle during S phase, with the appearance of one procentriole orthogonally to each of the two mother centrioles. Procentrioles then elongate during the following G2 phase of the cell cycle, acquiring the inner scaffold protein POC5 that is critical for the formation of the central and distal parts of the nascent procentriole (*Azimzadeh et al., 2009*). We followed endogenous WDR90 localization across the cell cycle by analyzing synchronized human RPE1 cells fixed at given time points and stained for either Centrin or HsSAS-6, both early protein marker of duplicating centrioles (*Azimzadeh et al., 2009*; *Strnad et al., 2007*; *Figure 3A–F* and *Figure 3—figure supplement 1A, B*). We found that while Centrin and HsSAS-6 are recruited as expected early on during procentriole formation in S phase (22 hr) (*Strnad et al., 2007*), WDR90 starts appearing only in early G2 when procentriole elongation starts (24 hr) (*Figure 3A–F*). Signal intensity analysis over the cell cycle further demonstrates that WDR90 appears on procentrioles in early G2 and reaches full incorporation by the end of G2 (*Figure 3G,H*), similarly to the reported incorporation of the inner scaffold protein POC5 (*Azimzadeh et al., 2009*).

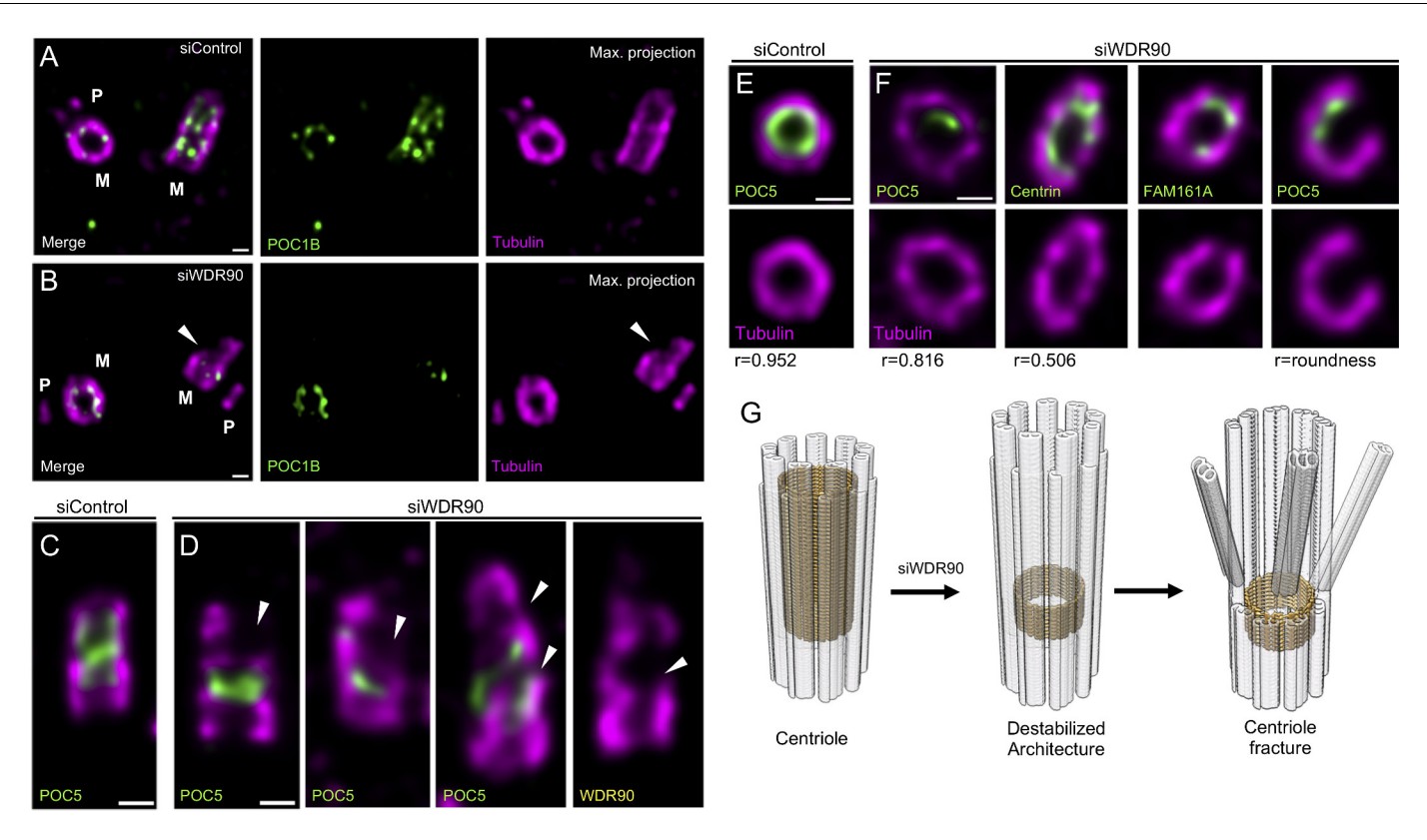

**Figure 5.** WDR90 is important for centriole architecture integrity (see also *Figure 5—figure supplement 1*, *Videos 1* and *2*). (A, B) Expanded centrioles from S-phase U2OS cells treated with control (A) or *wdr90* siRNA (B), stained for tubulin (magenta) and POC1B (green). White arrowhead: broken microtubule wall of the mature centriole. P: procentriole, M: mature centriole. Scale bars: 100 nm. (C, D) Expanded centrioles from U2OS cells treated with control (C) or *wdr90* siRNA (D), stained for tubulin (magenta) and POC5 (green) or WDR90 (yellow), displaying microtubule wall fractures (white arrowheads), lateral view. Scale bars: 100 nm. (E, F) Top views of expanded centrioles from U2OS cells treated with control (E) or *wdr90* siRNA (F) stained as specified above. Note the loss of roundness of centrioles treated with *wdr90* siRNA. Scale bars: 100 nm. (G) Model of WDR90 function holding microtubule triplets in the central core region of centrioles.

The online version of this article includes the following figure supplement(s) for figure 5:

**Figure supplement 1.** WDR90 depletion leads to severe centriolar structure defects.

In addition, we noticed that besides its centriolar distribution, WDR90 localizes also to centriolar satellites, which are macromolecular assemblies of centrosomal proteins scaffolded by the protein PCM1 and involved in centrosomal homeostasis (*Drew et al., 2017*; *Odabasi et al., 2020*; *Figure 3—figure supplement 1C–D*). Thus, we tested whether WDR90 satellite localization depends on the satellite protein PCM1 by depleting PCM1 using siRNA and assessing WDR90 distribution. We found that in absence of PCM1, WDR90 is solely found at centrioles (*Figure 3—figure supplement 1E–H*), demonstrating that WDR90 satellite localization is PCM1-dependent.

Altogether, these data establish that WDR90 is a centriolar and satellite protein that is recruited to centrioles in the G2-phase of the cell cycle, during procentriole elongation and central core/distal formation, similarly to the recruitment of the inner scaffold protein POC5.

## WDR90 is important to recruit Centrin and POC5

To better understand the function of WDR90, we analyzed cycling human cells depleted for WDR90 using siRNA and co-labeled WDR90 with the early centriolar marker Centrin. As previously shown (*Hamel et al., 2017*), WDR90 siRNA-treated cells showed significantly reduced WDR90 levels at

centrosomes in comparison to control cells (*Figure 3—figure supplement 2A,C*). Moreover, we observed an asymmetry in signal reduction at centrioles in WDR90-depleted cells, with only one of two Centrin-positive centrioles still associated with WDR90 in G1 and early S-phase (69% compared to 10% in controls) and one of four Centrin-positive centrioles in S/G2/M cells (77% compared to 0% in controls, *Figure 3—figure supplement 2B*). As the four Centrin-positive dots indicate duplicated centrioles, this result suggests that the loss of WDR90 does not result from a duplication failure (*Figure 3—figure supplement 2B*). We postulate therefore that the remaining WDR90 signal possibly corresponds to the mother centriole and that the daughter has been depleted from WDR90 (*Figure 3—figure supplement 2E*), similarly to what has been observed for the protein POC5 (*Azimzadeh et al., 2009*). We further conclude that WDR90 is stably incorporated into centrioles, in agreement with its possible structural role.

We also noted that the intensity of the Centrin and POC5 signals were markedly reduced upon WDR90 siRNA treatment (*Figure 3—figure supplement 2D–K*). Indeed, we found that only 39% of WDR90-depleted cells displayed 2 POC5 dots in G1 (negative for HsSAS-6 signal) in contrast to the 86% of control cells with 2 POC5 dots (*Figure 3—figure supplement 2H*). Moreover, 68% of control cells had 2 to 4 POC5 dots in S/G2/M (associated with 2 HsSAS-6 dots) in contrast to 29% in WDR90-depleted condition (*Figure 3—figure supplement 2H*). The HsSAS-6 signal was not affected in WDR90-depleted cells, confirming that initiation of the centriole duplication process is not impaired under this condition (*Figure 3—figure supplement 2G,J,L*). Similarly, the fluorescence intensity of the distal centriole cap protein CP110 was not changed under WDR90-depletion in contrast to the Centrin signal reduction (*Figure 3—figure supplement 2M–O*).

To ascertain the specificity of this phenotype, we generated a stable cell line expressing a siRNA-resistant version of WDR90 fused to GFP in its N-terminus (GFP-WDR90RR) upon doxycycline induction. We found that expression of GFP-WDR90RR restores partially the Centrin and POC5 signals at centrioles (*Figure 3I–L*).

Taken together, these results indicate that the depletion of WDR90 leads to a decrease in Centrin and POC5 localization at centrioles but does not affect the initiation of centriole duplication nor the recruitment of the distal cap protein CP110.

## WDR90 depletion leads to a loss of inner scaffold components and to centriole fracture

To investigate the structural role of POC16/WDR90 proteins on centrioles, we initially turned to the previously studied *Chlamydomonas reinhardtii poc16m504* and *poc16m55* mutants (*Hamel et al., 2017*; *Li et al., 2016*). However, after backcrossing these two strains with a wild-type strain (CC-124), it was found that the *poc16* mutation is unlinked to the motility phenotype of *poc16m555* and unlinked to the ciliary assembly defect of *poc16m504* previously reported (personal communication from Prof. Susan Dutcher, Washington University in St. Louis). Further genetic characterization will be needed to study the phenotypes associated with *poc16* mutations.

Therefore, we decided to analyze WDR90 phenotype in human cells and asked whether WDR90 depletion might lead to a loss of inner scaffold components as well as to a centriole architecture destabilization. We tested this hypothesis by analyzing centrioles from WDR90-depleted U2OS cells using U-ExM (*Figure 4*). As expected, we observed a strong reduction of WDR90 at centrioles, with a reminiscent asymmetrical signal in one of the two mature centrioles (*Figure 4A,B*). Unexpectedly, we found that WDR90-depleted centrioles exhibited a slight tubulin length increase (502 nm +/- 65 compared to 434 nm +/- 58 in controls), potentially indicative of a defect in centriole length regulation (*Figure 4C*). In contrast, despite a slight decrease at the level of the central core, we did not observe, in neither of the conditions, any significant difference in centriole diameter at the proximal and very distal regions (*Figure 4D*).

A key prediction is that the inner scaffold is connected to the microtubule wall through the stem structure that may contain WDR90. To test this, we next analyzed whether the localization of the four described inner scaffold components POC1B, FAM161A, POC5 and Centrin would be affected in WDR90-depleted cells. We found that the localization of these four proteins in the central core region of centrioles was markedly altered in WDR90-depleted daughter centrioles (*Figure 4E,F*) using CEP164 to label the mother centriole (*Figure 4—figure supplement 1A–C*). Instead of covering ~60% of the entire centriolar lumen, we only observed a ~ 20% remaining belt, positive for inner scaffold components at the proximal extremity of the core region (*Figure 4E–G* and *Figure 4—*

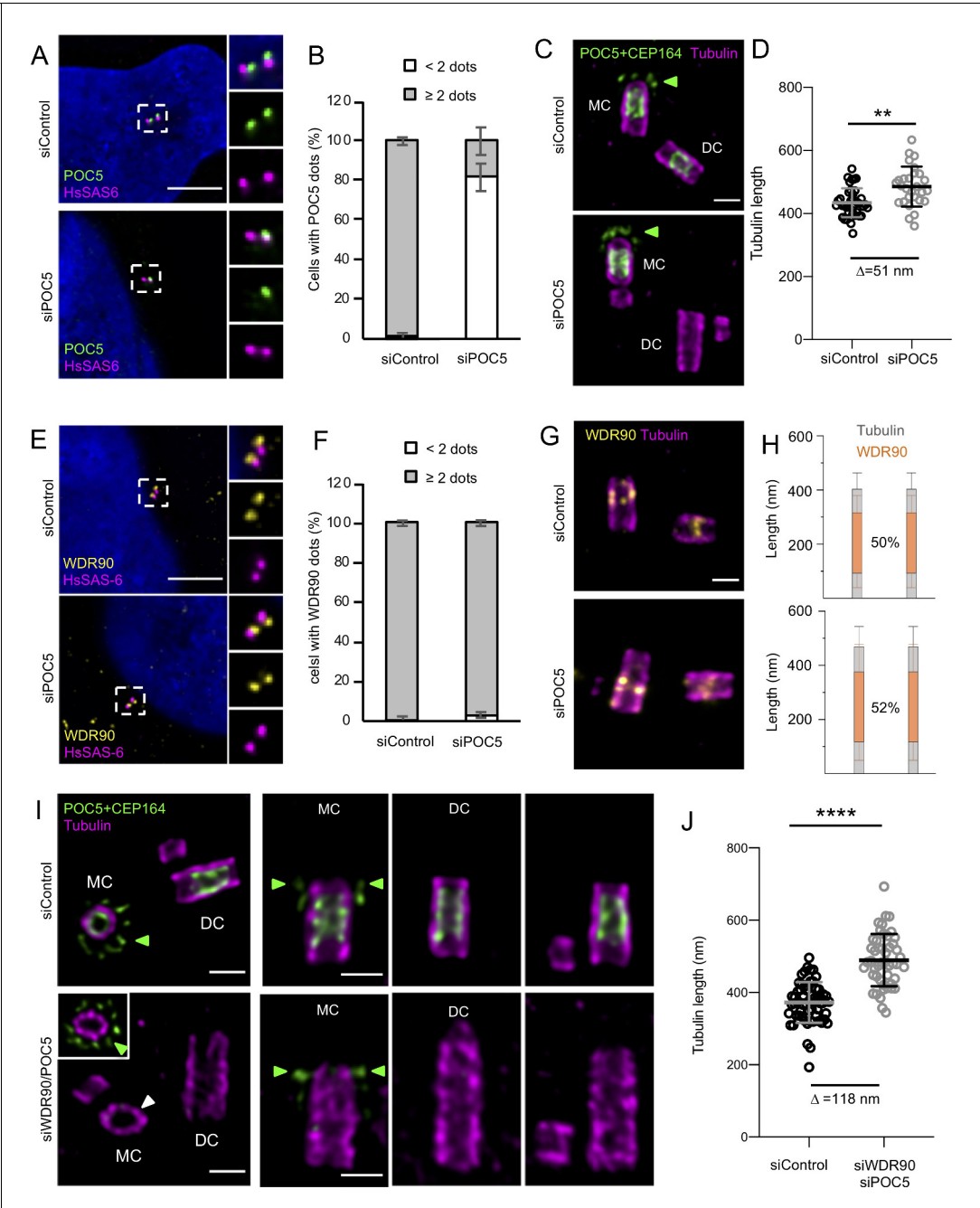

**Figure 6.** POC5 and WDR90 are important for proper centriole architecture. (See also *Figure 6—figure supplements 1* and *2*) (**A**) Human U2OS cell treated with either control or *poc5* siRNA and stained for POC5 (green) and HsSAS-6 (magenta). DNA is in blue. Dotted white squares indicate insets. Scale bar: 5 μm. (**B**) Percentage of cells with the following number of POC5 dots per cell based on A, n = 50 cells/condition from three independent experiments. Average +/- SD: refer to *Figure 6—source data 1*. Fisher's exact test p<0.0001. (**C**) Expanded centrioles from U2OS treated with either control or *poc5* siRNA stained for tubulin (magenta) and POC5+CEP164 (both in green. CEP164 is indicated by a green arrowhead). MC stands for mother centriole and DC for daughter centriole. Scale bar: 250 nm. (**D**) Tubulin length in nm, n = 30 centrioles/condition from two independent experiments. Average +/- SD: siControl = 434 nm +/- 45, siPOC5 = 485 nm +/- 64. Mann-Whitney p=0.0005. (**E**) Human U2OS cell treated with either control or *poc5* siRNA and stained for WDR90 (yellow) and HsSAS-6 (magenta). DNA is in blue. Dotted white squares indicate insets. Scale bar: 5 μm. (**F**) Percentage of cells with the following number of WDR90 dots per cell based on A, n = 50 cells/condition from three independent experiments. Average +/- SD: refer to *Figure 6—source data 2*. Fisher's exact test p=0.6328. (**G**) Expanded centrioles from U2OS treated with either control or *poc5* siRNA stained for tubulin (magenta) and WDR90 (yellow). Scale bar: 250 nm. (**H**) Average WDR90 length coverage in siControl or siPOC5. n = 30 centrioles/condition from two independent experiments. Average +/- SD: siControl = 50% +/- 21; siPOC5 = 52% +/- 23. (**I**) Expanded centrioles from U2OS treated with either control or *wdr90/poc5* siRNA stained for tubulin (magenta) and POC5+CEP164 (both in green, CEP164 is indicated by a green

*Figure 6 continued on next page*

*Figure 6 continued*

arrowhead). MC stands for mother centriole and DC for daughter centriole. Inset shows a distal position of the mother centriole were CEP164 signal is visible (green arrowheads). White arrowhead indicates a loss of centriolar roundness. Scale bars: 250 nm. (**J**) Tubulin length in nm, n = 50 centrioles/condition from two independent experiments. Average +/- SD: siControl = 372 nm +/- 56, siWDR90/POC5 = 490 nm +/- 72. Unpaired t test ****p<0.0001.

The online version of this article includes the following source data and figure supplement(s) for figure 6:

**Source data 1.** Percentage of cells with the following number POC5 dots/cell in siControl and siPOC5 conditions.
**Source data 2.** Percentage of cells with the following number WDR90 dots/cell in siControl and siPOC5 conditions.
**Figure supplement 1.** Characterization of POC5 and POC5/WDR90 depletion.
**Figure supplement 1—source data 1.** Length of centriole in metaphase and at the end of mitosis in siControl and siPOC5 conditions.
**Figure supplement 1—source data 2.** Percentage of cells with the following number POC5 dots/cell in siControl and siWDR90/POC5 conditions.
**Figure supplement 1—source data 3.** Percentage of cells with the following number WDR90 dots/cell in siControl and siWDR90/POC5 conditions.
**Figure supplement 2.** Loss of the inner scaffold components WDR90 and POC5 leads to centriole breakage.

*figure supplement 1D,E*), suggesting that their initial recruitment may not be entirely affected. Another possibility would be that incomplete depletion of WDR90 allows for partial localization of inner scaffold components. It should also be noted that Centrin, which displays a central core and an additional distal tip decoration (*Le Guennec et al., 2020*), was affected specifically in its inner core distribution (*Figure 4E* white arrow, *Figure 4—figure supplement 1D,E*).

The discovery of the inner scaffold within the centriole led to the hypothesis that this structure is important for microtubule triplet stability and thus overall centriole integrity (*Le Guennec et al., 2020*). In line with this hypothesis, we found that upon WDR90 depletion, 10% of cells had their centriolar microtubule wall broken, indicative of microtubule triplets fracture and loss of centriole integrity (15 out of 150 centrioles, *Figure 5*, *Videos 1* and *2*). The break occurred mainly above the remaining belt of inner scaffold components (*Figure 5A–D*), possibly reflecting a weakened microtubule wall in the central and distal region of the centriole. We also noticed that the perfect cylindrical shape (defined as roundness) of the centriolar microtubule wall was affected with clear ovoid-shaped or opened centrioles seen from near-perfect top view oriented centrioles (*Figure 5E,F* and *Figure 5—figure supplement 1*, 95% of depleted centrioles in top view are affected), illustrating that loss of WDR90 and the inner scaffold leads to disturbance of the characteristic centriolar architecture.

To assess whether WDR90 stability phenotype correlates solely with disturbance of inner scaffold proteins, we analyzed the distribution of the centriolar proteins FOP1 and CEP135 (BLD10) as well as glutamylation (PolyE), all known to be important for centriole stability (*Bayless et al., 2012*; *Bayless et al., 2016*; *Bobinnec et al., 1998*; *Lin et al., 2013*; *Matsuura et al., 2004*). While CEP135 and glutamylation were not altered in WDR90-depleted cells (*Figure 4—figure supplement 1F–K*), we found that FOP1 distribution was slightly disturbed at centrioles (*Figure 4—figure supplement 1L–N*) but still present, reinforcing our interpretation that the centriole breakage is probably due to the loss of the inner scaffold components.

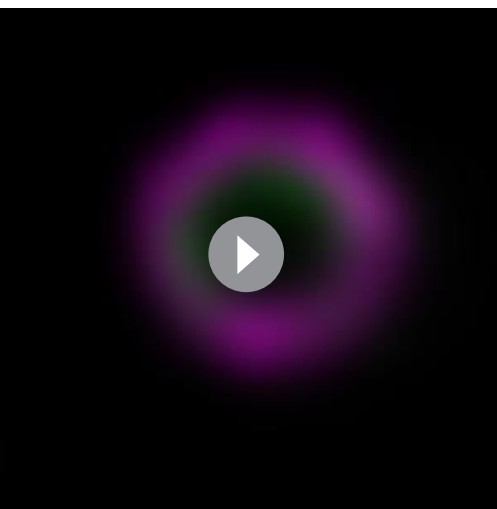

**Video 1.** U-ExM expanded control centrioles. Top viewed expanded centriole from U2OS cell treated with control siRNA and stained for tubulin (magenta) and POC5 (green). Z-stack acquired every 0.12 μm from the proximal to distal end of the centriole.
https://elifesciences.org/articles/57205#video1

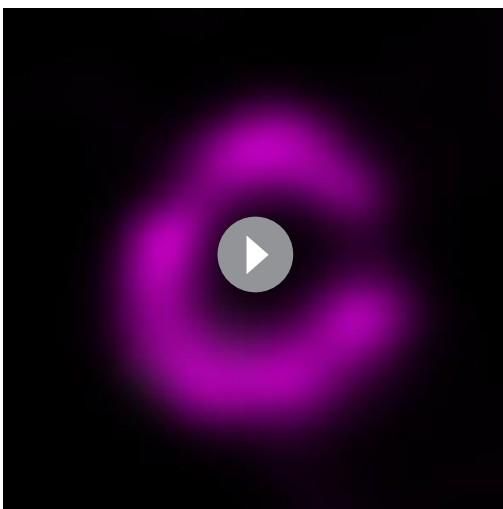

**Video 2.** U-ExM expanded centrioles depleted of WDR90. Top viewed expanded centriole from U2OS cell treated with *wdr90* siRNA and stained for tubulin (magenta) and POC5 (green). Z-stack acquired every 0.12 µm from the proximal to distal end of the centriole.

https://elifesciences.org/articles/57205#video2

## WDR90/POC5 co-depletion enhances centriole architecture abnormalities

As the inner scaffold connects the microtubule triplet together, we wondered whether the remaining belt seen in WDR90 depleted cells could limit the phenotype of centriolar breakage. To test this hypothesis, we decided to co-deplete WDR90 with the inner scaffold protein POC5. We first depleted POC5 alone using previously described siRNA (*Figure 6A, B*; *Azimzadeh et al., 2009*). Consistently with WDR90 depletion, we found that the removal of the inner scaffold POC5, which occurs mainly at daughter centrioles (*Figure 6—figure supplement 1A*), led to a slight centriole elongation (*Figure 6C,D*) and resulted in 10% of broken centrioles (*Figure 6—figure supplement 2A,B*; 4 out of 46 centrioles). We also confirmed that POC5 depletion leads to shorter procentrioles in metaphase as previously reported (*Azimzadeh et al., 2009*) but then become over elongated just after mitosis (*Figure 6—figure supplement 1B,C*). We next assessed whether POC5 depletion would impair WDR90 distribution; however, we found this not to be the case, as WDR90 localization is not affected at centrioles upon POC5 depletion (*Figure 6E–H* and *Figure 6—figure supplement 1D*). This result therefore indicates that WDR90 is upstream of POC5.

We next capitalize on this efficient POC5 depletion to co-deplete POC5 together with WDR90 (*Figure 6—figure supplement 1E–J*). We found that the double siRNA led to a strong decrease of cell number as compared to WDR90 depletion alone, suggesting either an increase of cell mortality or a defect in cell cycle progression (*Figure 6—figure supplement 1J*). As expected, we found that the remaining POC5 belt found in WDR90-depleted centrioles was completely removed (*Figure 6I*). Moreover, centrioles appeared even further elongated under these conditions, indicating that the complete removal of POC5 further enhances the WDR90 phenotype (*Figure 6I,J*). Structurally, we noticed beside the elongated centrioles about 30%, of abnormal centrioles in WDR90/POC5 depleted cells (*Figure 6—figure supplement 2A,C,D*; 70 out of 260 centrioles), ranging from very short centrioles that seem to lack the entire core/distal region as well as centrioles with broken microtubule blades. We also noted a loss of centriole roundness (*Figure 6I*, white arrow). Overall, these phenotypes support our prediction that depletion of inner scaffolds component strongly impairs centriole integrity.

Collectively, we demonstrate that WDR90 is crucial to ensure inner core protein localization within the centriole core, as well as to maintain the microtubule wall integrity and the overall centriole roundness and stability (*Figure 5G*).

## Discussion

What maintains centriole barrel stability and roundness is a fundamental open question. Centrioles are microtubule barrel structures held together by the A-C linker at their proximal region and a recently discovered inner scaffold in the central/distal region (*Le Guennec et al., 2020*). The presence of such an extended scaffold covering 70% of the centriolar length has led to the hypothesis that this structure is important for maintaining centriole integrity (*Le Guennec et al., 2020*). Our work demonstrates that POC16/WDR90 family proteins constitute an evolutionary conserved central core microtubule triplet component that is essential for maintaining the inner centriolar scaffold components in human centrioles. The depletion of WDR90 leads to centriolar defects and impairment of microtubule triplets organization resulting in the loss of the canonical circular shape

of centrioles. We also found that this overall destabilization of the centriole can lead to microtubule triplet breakage. Whether this phenotype arises as a consequence of the loss of the inner scaffold or due to the destabilization of the inner junction of the microtubule triplet is still an opened question that should be addressed in the future. Moreover, although unlikely, we cannot exclude that fragile centrioles such as the ones found in WDR90-depleted cells could be affected and further distorted by the technique of expansion microscopy.

We also demonstrate using expansion microscopy that POC16/WDR90 is a component of the microtubule triplet restricted to the central core region. In addition and based on the sequence and structural similarity to the DUF667 domain of FAP20 that composes the inner junction in flagella, we propose that POC16/WDR90 localizes at the inner junction of the A and B microtubule of the centriolar microtubule triplet. The fact that WDR90 localization is restricted to the central core region led us to hypothesize that another protein, possibly FAP20 as it has been previously reported at centrioles (*Yanagisawa et al., 2014*), could mediate the inner junction between A- and B-microtubule in the proximal region of the centriole. Moreover, in POC16/WDR90 proteins, the DUF667 domain is followed by a WD40 domain sharing a similarity with the flagellar inner B-microtubule protein FAP52/WDR16 (*Owa et al., 2019*) leading us to propose that the WD40 domains of POC16/WDR90 might also be located inside the B-microtubule of the triplet. However, whether this is the case remains to be addressed in future studies. In addition, WDR90 is potentially not the only protein that forms the inner junction. Indeed, we and others also previously show that FAM161A (*Le Guennec et al., 2020*; *Zach et al., 2012*), similarly to WDR90, is a microtubule-binding protein close to the inner microtubule wall of the centriole, raising the possibility that both might compose the stem and link the microtubule triplets to the inner scaffold. It will be interesting in the future to study whether these two proteins interact.

Our work further establishes that WDR90 is recruited to centrioles in G2 phase of the cell cycle concomitant with centriole elongation and inner central core assembly. We found that WDR90 depletion does not impair centriole duplication nor microtubule wall assembly, as noted by the presence of the proximal marker HsSAS-6 and the distal cap CP110. In stark contrast, WDR90 depletion leads to a strong reduction of inner scaffold components at centrioles, as well as some centriole destabilization.

Although several examples of centriole integrity loss have been demonstrated in the past, the molecular mechanisms of centriole disruption are not understood. For instance, Delta- and Epsilon-tubulin mutants have been shown in several model organisms to affect centriole integrity (*Dutcher et al., 2002*; *Dutcher and Trabuco, 1998*; *Garreau de Loubresse et al., 2001*; *O'Toole et al., 2003*) with notably in human cells where Delta- and Epsilon-tubulin null mutant cells were shown to lack microtubule triplets and have thus unstable centrioles that do not persist to the next cell cycle (*Wang et al., 2017*). Remarkably, these centrioles can elongate with a proper recruitment of the cartwheel component HsSAS-6 and the distal marker CP110 but fails to recruit POC5, a result that is similar to our findings with WDR90-depleted cells. As Delta- and Epsilon-tubulin null human mutant cells can solely assemble microtubule singlets (*Wang et al., 2017*), we speculate that WDR90 might not be recruited in these centrioles, as the A- and B-microtubule inner junction would be missing. As a consequence, the inner scaffold proteins may not be recruited, as already shown for POC5, leading to the observed futile cycle of centriole formation and disintegration (*Wang et al., 2017*). It would therefore be interesting to study the presence of WDR90 in these null mutants as well as the other components of the inner scaffold in the future.

Our work also showed that WDR90 as well as POC5 depletion affects centriole length in human cells. Altogether, these results emphasize the role of these two proteins in overall centriole length regulation and suggest an unexpected role of the inner scaffold structure in centriole length control. It would be of great interest to understand if and how the absence of the inner scaffold can affect the length of the centriole without affecting distal markers such as CP110, which remains unchanged in our experiments. It is very likely that the concomitant elongation of the centriole with the appearance of inner scaffold components in G2 can act on the final length of this organelle.

Given the importance of centriole integrity in enabling the proper execution of several diverse cellular processes, our work provides new fundamental insights into the architecture of the centriole, establishing a structural basis for centriole stability and the severe phenotypes that arise when lost.

# Materials and methods

**Key resources table**

| Reagent type (species) or resource | Designation | Source or reference | Identifiers | Additional information |
|---|---|---|---|---|
| Strain, strain background (*Chlamydomonas reinhardtii*) | WT | *Chlamydomonas* Resource Center | cMJ030 | Wild-type |
| Strain, strain background (*Paramecium tetraurelia*) | 7S | *Beisson et al., 2010* | doi:10.1101/pdb.prot5364 | |
| Cell line (*Homo sapiens*) | U2OS | *Habedanck et al., 2005* | PMID:16244668 | |
| Cell line (*Homo sapiens*) | RPE-1 p53- | *Wang et al., 2015* | PMID:26609813 | |
| Cell line (*Homo sapiens*) | U2OS:GFP-WDR90RR | This paper | p. 19 of the manuscript (Material and methods) | Episomal, puromycine selected, doxycycline-inducible |
| Transfected construct (*Homo sapiens*) | GFP-WDR90RR | This paper | pEBTet-GFP-WDR90RR(FL) p. 20 of the manuscript (Material and methods) | WDR90RR DNA template from *Hamel et al., 2017* |
| Transfected construct (*Homo sapiens*) | GFP-WDR90(1-225)RR | This paper | pEBTet-GFP-WDR90RR(1-225) p. 20 of the manuscript (Material and methods) | WDR90RR DNA template from *Hamel et al., 2017* |
| Transfected construct (*Homo sapiens*) | GFP-WDR90 | This paper | Genebank sequence NP_660337, pEGFP-WDR90 | RT-PCR from human RPE-1 cells, cloned into modified pEGFP-C1 vector using *AscI* and *PacI* restriction sites |
| Transfected construct (*Chlamydomonas reinhardtii*) | POC16 (1-295) | This paper | pXLG-POC16(1-295), described p.20 of the manuscript in the Material and methods section. | POC16 sequence synthetized by GeneArt using the *E. coli* codon usage (described in *Hamel et al., 2017*) cloned into pXLG vector using NotI and BamHI restriction sites |
| Biological sample (*Chlamydomonas reinhardtii*) | Isolated basal bodies | *Klena et al., 2018* | PMID:30295659 | |
| Biological sample (*Sus scrofa*) | Tubulin | Cytoskeleton | Cat. #: T240 | Isolated from brain, used for electron microscopy |
| Biological sample (*Bovine taurus*) | Tubulin | Centro de Investigastiones Biologicas, Madrid, Spain | | Isolated from brain, used for pelleting assay |
| Antibody | Tubulin AA345 (mouse monoclonal) | *Le Guennec et al., 2020* | PMID:32110738 | U-ExM Isolated Basal Bodies (1:500) U-ExM in cells (1:250) |
| Antibody | Alpha-Tubulin AA344 (mouse monoclonal) | *Le Guennec et al., 2020* | PMID:32110738 | U-ExM in cells (1:250) |
| Antibody | POC16 (rabbit polyclonal) | *Hamel et al., 2017* | PMID:28781053 | U-ExM (1:100) |
| Antibody | POB15 (rabbit polyclonal) | *Hamel et al., 2017* | PMID:28781053 | U-ExM (1:100) |
| Antibody | WDR90 (rabbit polyclonal) | NovusBio | Cat. #: NBP2-31888 | U-ExM (1:100) IF (1:250) |

*Continued on next page*

*Continued*

| Reagent type (species) or resource | Designation | Source or reference | Identifiers | Additional information |
|---|---|---|---|---|
| Antibody | POC1B (rabbit polyclonal) | ThermoFisher | Cat. #: PA5-24495 | U-ExM (1:250) |
| Antibody | POC5 (rabbit polyclonal) | Bethyl | Cat. #: A303-341A | U-ExM (1:250) IF (1:500) |
| Antibody | FAM161A (rabbit polyclonal) | *Le Guennec et al., 2020* | PMID:32110738 | U-ExM (1:250) |
| Antibody | Centrin (mouse monoclonal, 20H5) | Merck Millipore | Cat. #: 04–1624 | U-ExM (1:250) IF (1:500) |
| Antibody | DM1A Tubulin (mouse monoclonal) | Abcam | Cat. #: ab7291 | IF (1:1000) |
| Antibody | HsSAS-6 (mouse monoclonal) | Santa Cruz Biotechnology | Cat. #: sc-81431 | IF (1:100) |
| Antibody | PCM1 (rabbit polyclonal) | Santa Cruz Biotechnology | Cat. #: sc-67204 | IF (1:500) |
| Antibody | CP110 (rabbit polyclonal) | Proteintech | Cat. #: 12780–1 | IF (1:500) |
| Antibody | GFP (mouse monoclonal) | Abcam | Cat. #: ab1218 | IF (1:500) |
| Antibody | mCherry (rabbit polyclonal) | Abcam | Cat. #: ab167453 | IF (1:500) |
| Antibody | ptPOC16 (rabbit polyclonal) | This study | described p.27 of the manuscript in the Supplemental Methods section. | IF (1:50) |
| Antibody | Tubulin 1D5 (mouse mono clonal) | *Beisson et al., 2010* | | IF (1:10) |
| Antibody | Alexa 488 anti-rabbit IgG (goat) | ThermoFisher | Cat. #: A11008 | U-ExM (1:400) IF (1:1000) |
| Antibody | Alexa 568 anti-mouse IgG (goat) | ThermoFisher | Cat. #: A11004 | U-ExM (1:400) IF (1:1000) |
| Recombinant DNA reagent | pEBTet-EGFP-GW | Gift from the Gönczy lab | Na. | |
| Recombinant DNA reagent | pENTR-Age-AGT | Gift from the Gönczy lab | Na. | |
| Recombinant DNA reagent | pEGFP-C1 | Clontech | | |
| Sequence-based reagent | siRNA Control | ThermoFisher | AM4642 | Silencer select |
| Sequence-based reagent | siRNA targeting *wdr90* gene | ThermoFisher | S47097 | Silencer select |
| Sequence-based reagent | siRNA targeting *pcm1* gene | ThermoFisher | ADCSU9L | Silencer select |
| Peptide, recombinant protein | POC16(1-295) | This paper | Uniprot A8JAN3 | Purified from bacteria |
| Peptide, recombinant protein | WDR90(1-225) | This paper | Uniprot Q96KV7 | Purified from bacteria |
| Peptide, recombinant protein | drPOC16(1-243) | This paper | Uniprot F1RA29 | Purified from bacteria |
| Peptide, recombinant protein | btPOC16(1-224) | This paper | Uniref UPI000572B175 | Purified from bacteria |
| Peptide, recombinant protein | ptPOC16(2-210) | This paper | Uniprot A0DK60 | Purified from bacteria |

*Continued on next page*

*Continued*

| Reagent type (species) or resource | Designation | Source or reference | Identifiers | Additional information |
|---|---|---|---|---|
| Peptide, recombinant protein | xtPOC16(1-245) | This paper | Uniref UPI0008473371 | Purified from bacteria |
| Peptide, recombinant protein | rnPOC16(54-282) | This paper | Uniref UPI0008473371 | Purified from bacteria |
| Commercial assay or kit | Lipofectamine 3000 Transfection kit | LifeTechnology | Cat. #: L3000015 | |
| Commercial assay or kit | Lipofectamine RNAi max kit | LifeTechnology | Cat. #: 13778150 | |
| Commercial assay or kit | Click-EdU-Alexa647 FACS kit | Carl Roth | Cat. #: 7783.1 | |
| Commercial assay or kit | DAPCO Mounting medium | Abcam | Cat. #: ab188804 | |
| Commercial assay or kit | Affi-Gel 10 | Bio-Rad | Cat. #:153–6099 | |
| Chemical compound, drug | Formaldehyde 36.5–38% | Sigma | Cat. #: F8775 | |
| Chemical compound, drug | Acrylamide 40% | Sigma | Cat. #: A4058 | |
| Chemical compound, drug | N,N'-methyl bisacrylamide 2% | Sigma | Cat. #: M1533 | |
| Chemical compound, drug | Sodium acrylate 97–99% | Sigma | Cat. #: 408220 | |
| Chemical compound, drug | Ammonium persulfate | ThermoFisher | Cat. #: 17874 | |
| Chemical compound, drug | Tetramethylethyldiamine | ThermoFisher | Cat. #: 17919 | |
| Chemical compound, drug | Poly-D-Lysine 1 mg/mL | Gibco | Cat. #: A3890401 | |
| Chemical compound, drug | Taxol/Paclitaxel | Sigma-Aldrich | Cat. #: T7191 | |
| Chemical compound, drug | Coomassie staining | Biotium | Cat. #: 21003 | |
| Chemical compound, drug | Propidium Iodide | Sigma | Cat. #: 81845 | |
| Chemical compound, drug | Rnase | Roche | Cat. #: 11119915001 | |
| Software, algorithm | ImageJ/FiJi | *Schindelin et al., 2012* | doi:10.1038/nmeth.2019 | |
| Software, algorithm | CentrioleJ pluggin | *Guichard et al., 2013* | DOI:10.1016/j.cub.2013.06.061 | |
| Software, algorithm | UnwarpJ pluggin | *Sorzano et al., 2005* | DOI:10.1109/TBME.2005.844030 | |
| Software, algorithm | GraphPadPrism7 | GraphPad Software | 7.0 | |
| Software, algorithm | Phyre2 | *Kelley and Sternberg, 2009* | DOI:10.1038/nprot.2015.053 | |
| Software, algorithm | UCSF Chimera | *Pettersen et al., 2004* | DOI:10.1002/jcc.20084 | |
| Other | Zeiss LSM700 microscope | Zeiss | | |
| Other | Leica TCS SP8 microscope | Leica | | Expansion microscopy |
| Other | Leica Thunder inverted microscope | Leica | | |
| Other | Tecnai G2 Sphera microscope | Thermofisher | | Negative stain and cryo-EM |

## Method details

### Human cell lines

Human U2OS and RPE1 p53- cells (gift from Meng-Fu Bryan Tsou) were cultured similarly to *Hamel et al., 2017*. This cell lines have been authenticated by Microsynth. Cells were grown in DMEM supplemented with GlutaMAX (Life Technology), 10% tetracycline-negative fetal calf serum (life technology), penicillin and streptomycin (100 μg/ml). Cell lines were regularly tested for mycoplasma contamination using the Mycoplasma detection Kit-Quick Test (biotool.com, cat: B39032).

To generate inducible episomal U2OS:GFP-WDR90RR cell line, U2OS cells were transfected using Lipofectamine 3000 (Life Technology). Transfected cells were selected for 6 days using 1 μg/mL puromycin starting day 2 after transfection. Selected cells were amplified and frozen. For further experiments, U2OS:GFP-WDR90 cell line was grown in the medium specified above supplemented with 1 μg/mL puromycin.

### Cloning and protein purification

The constructs encompassing the predicted DUF667 domain of POC16 (Uniprot: A8JAN3), WDR90 (Uniprot: Q96KV7), drPOC16 (Uniprot: F1RA29), btPOC16 (Uniref: UPI000572B175), ptPOC16 (Uniprot: A0DK60), xtPOC16 (Uniref: UPI0008473371) and rnPOC16 (Uniref UPI0008473371) were cloned into a pET-based expression vector via Gibson assembly (*Gibson et al., 2009*).

All recombinant proteins contained a N-terminal thioredoxin (TrxA) tag, used to enhance the expression level and the solubility of the target protein, followed by a 6xHis tag and a 3C cleavage site.

Protein expression was carried out in *E. coli* BL21 (DE3) competent cells grown in LB media at 37°C to $OD_{600}$ = 0.6 and induced for 16 hr at 20°C with 1 mM IPTG. Cells were subsequently resuspended in lysis buffer (50 mM Hepes pH 8, 500 mM NaCl, 10% v/v glycerol, 10 mM imidazole pH 8, 5 mM β-mercaptoethanol) supplemented with DNase I (Sigma), complete EDTA-free protease inhibitor cocktail (Roche) and lysed by sonication. The supernatant was clarified by centrifugation (18,000 rpm, 4 °C, 45 min), filtered and loaded onto a HisTrap HP 5 ml column (GE Healthcare). After extensive washes with wash buffer (50 mM Hepes pH 8, 500 mM NaCl, 10% v/v glycerol, 20 mM imidazole pH 8, 5 mM β-mercaptoethanol), the bound protein was eluted in the wash buffer supplemented with 400 mM imidazole. For POC16, WDR90, drPOC16 and xtPOC16, a 10 to 400 mM imidazole gradient was required to successfully detach the protein from the column.

The protein-containing fractions were pooled together and dialysed against the lysis buffer at 4 °C for 48 hr in the presence of the 6xHis-3C protease. The tag-free protein was reapplied onto a HisTrap HP 5 ml column (GE Healthcare) to separate the cleaved product from the respective tags and potentially uncleaved protein. The processed proteins were concentrated and further purified by size exclusion chromatography (Superdex-75 16/60, GE Healthcare) in running buffer (20 mM Tris pH 7.5, 150 mM NaCl, 2 mM DTT). Protein were analysed by Coomasie stained SDS-PAGE and the protein-containing fractions were pooled, concentrated and flash-frozen for storage at −80 °C. All protein concentrations were estimated by UV absorbance at 280 nm.

### Microtubule binding assay

Taxol-stabilized microtubules (MTs) were assembled in BRB80 buffer (80 mM PIPES-KOH pH6.8, 1 mM $MgCl_2$, 1 mM EGTA) from pure bovine brain tubulin at 1 mg/mL (Centro de Investigaciones Biológicas, Madrid, Spain). 50 μL of stabilized MTs were incubated with 20 μL of protein at 1 mg/mL for 2 hr at room temperature. After centrifugation on a taxol-glycerol cushion (8000 rpm, 30°C, 20 min) the supernatant and the pellet were analyzed by Coomasie stained SDS-PAGE gels. As a control, MTs alone and each protein alone were processed the same way.

### Tubulin-binding assay

Tubulin at 10 μM was incubated with a slight molar ratio excess of each protein construct (around 15 μM) in MES buffer for 15 min on ice. After centrifugation at 13,000 x g at 4°C for 20 min, the supernatant and the pellet were analyzed by Coomasie stained SDS-PAGE.

## In vitro microtubules decoration and imaging

For simple decoration, Taxol-stabilized microtubules were nucleated as described (*Schmidt-Cernohorska et al., 2019*) and subsequently exposed to recombinant WDR90-N(1-225) in a 1:1 molar ratio for 30 min at room temperature. 5 μL of protein complexes solution were blotted on carbon square 300 mesh grids (EMS) and stained with Uranyl Acetate (2%) for 3 then 30 s.

For double decoration, in vitro microtubules were incubated with WDR90-N(1-225) in a 1:1 molar ratio for 5 min at room temperature prior to addition of 2X free tubulin for 30 min at room temperature. Negatively stained grids were prepared as above. For cryo-microscopy, 4 μL of double decorated microtubule were deposited on a Lacey Carbon film grid (300 Mesh, EMS), blotted manually for 2 s and plunge into liquid ethane using an homemade plunger. Electron micrographs were acquired on a Tecnai G2 Sphera electron microscope (FEI Company) and analyzed using ImageJ.

## Cloning and transient overexpression in human cells

GFP-WDR90-N(1-225)RR and GFP-WDR90(FL)RR were cloned in the Gateway compatible vector pEBTet-eGFP-GW. Previously generated RNAi-resistant WDR90 DNA (*Hamel et al., 2017*) was used as template for PCR amplification. In brief, inserts were first subcloned in pENTR-Age-AGT using the restriction sites AgeI and XbaI. Second, a Gateway reaction was performed to generate the final expression plasmids pEBTet-GFP-WDR90-N(1-225)RR and pEBTer-GFP-WDR90(FL)RR, which were sequenced verified prior to transfection in human cells.

For transient expression, U2OS cells were transfected using Lipofectamine 3000 (Life Technology). Protein expression was induced using 1 μg/mL doxycycline for 48 hr and cells were processed for immunofluorescence analysis.

Cloning of the GFP-WDR90 construct used in *Figure 2* was done as follows: WDR90 was cloned by nested RT-PCR using total RNAs extracted from human RPE1 cells. Three different fragments corresponding to aa. 1–578, 579–1138, 1139–1748 of WDR90 (based on Genebank sequence NP_660337) were amplified and cloned separately using the pCR Blunt II Topo system (Thermo Fisher Scientific). The full coding sequence was then reconstituted in pCR Blunt II by two successive cloning steps using internal *Nru* I and *Sal* I, introduced in the PCR primers and designed in order not to modify WDR90 aa sequence. WDR90 coding sequence was then cloned into a modified pEGFP-C1 vector (Clontech) containing *Asc* I and *Pac* I restriction sites.

Cloning of POC16(1-295) into the pXLG vector was performed as followed: the POC16 sequence synthetized by GeneArt using the *E. coli* codon usage (described in *Hamel et al., 2017*) was cloned into pXLG vector using NotI and BamHI restriction sites.

## siRNA-mediated protein depletion

U2OS cells were plated onto coverslips in a 6-well plate at 200 000 cell/well 24 hr prior transfection.

For POC5 depletion, cells were transfected with 20 nM silencer select negative control siRNA1 (4390843, Thermo Fisher) and siPOC5 (sequence Sense siPOC5-1: 5' CAACAAAUUCUAGUCAUAC UU 3' and antisense: 5' GUAUGACUAGAAUUUGUUGCU 3', adapted from *Azimzadeh et al., 2009*) using Lipofectamine RNAimax (Thermo Fischer Scientific). Medium was changed 4 hr post-transfection and cells were analyzed 48 hr post-transfection.

For WDR90 depletion, cells were transfected with 10 nM of silencer select negative control siRNA1 and silencer select pre-designed siRNA s47097 using INTERFERin siRNA transfection reagent (Polyplus). After 48 hr, medium was changed and cells were analyzed 96 hr post-transfection.

For WDR90/POC5 depletion, cells were transfected with 10 nM of silencer select negative control siRNA1 and silencer select pre-designed siRNA s47097 using INTERFERin siRNA transfection reagent (polyplus). Medium was changed at 48 hr prior transfection and cells were subsequently transfected with 20 nM silencer select negative control siRNA1 and siPOC5 using INTERFERin siRNA transfection reagent (Polyplus). Cells were analyzed 48 hr after the second transfection.

In U2OS:GFP-WDR90(FL-RR) stable cell line, RNA-resistant protein expression was induced constantly for 96 hr using 1 μg/mL doxycycline.

## Immunofluorescence in human cells

Cells grown on a 15 mm glass coverslips (Menzel Glaser) were pre-extracted for 15 s in PBS supplemented with 0.5% triton prior to iced-cold methanol fixation for 7 min. Cells were washed in PBS then incubated for 1 hr in 1% bovine serum albumin (BSA) in PBS-T with primary antibodies against WDR90 (1:250, rabbit polyclonal, NovusBio NBP2-31888) (note that the WDR90 antibody also decorates the border of the cell, reminiscent to focal adhesion pattern), Centrin (1:500, mouse monoclonal, clone 20H5, 04–1624, Merck Millipore), POC5 (1:500, rabbit polyclonal, A303-341A, Bethyl) HsSAS-6 (1:100, mouse monoclonal, sc-81431, Santa Cruz Biotechnology), PCM1 (1:500, rabbit polyclonal, sc-67204, Santa Cruz Biotechnology), CP110 (1:500, rabbit polyclonal, 12780–1, Proteintech), GFP (1:500, mouse monoclonal, ab1218, Abcam), mCherry (1:500, rabbit polyclonal) or tubulin (1:500, mouse monocolonal, ab7291, Abcam). Coverslips were washed in PBS for 30 min prior to incubation with secondary antibodies (1:1000) for 1 hr at room temperature, washed again for 30 min in PBS and mounted in DAPCO mounting medium containing DAPI (Abcam). The following secondary antibodies were used: goat anti-rabbit Alexa Fluor 488 IgG H+L (1:400, A11008) and goat anti-mouse Alexa Fluor 568 IgG H+L (1:250, A11004) (Invitrogen, ThermoFisher).

Imaging was performed on a Zeiss LSM700 confocal microscope or on a Leica Thunder DMi8 microscope with a PlanApo 63x oil immersion objective (NA 1.4) and optical sections were acquired every 0.33 μm, then projected together using ImageJ.

## Ultrastructure Expansion Microscopy (U-ExM)

The following reagents were used in U-ExM experiments: formaldehyde (FA, 36.5–38%, F8775, SIGMA), acrylamide (AA, 40%, A4058, SIGMA), N,N'-methylenbisacrylamide (BIS, 2%, M1533, SIGMA), sodium acrylate (SA, 97–99%, 408220, SIGMA), ammonium persulfate (APS, 17874, ThermoFisher), tetramethylethylendiamine (TEMED, 17919, ThermoFisher), nuclease-free water (AM9937, Ambion-ThermoFisher) and poly-D-Lysine (A3890401, Gibco).

Monomer solution (MS) for one gel is composed of 25 μl of SA (stock solution at 38% (w/w) diluted with nuclease-free water), 12.5 μl of AA, 2.5 μl of BIS and 5 μl of 10X phosphate-buffered saline (PBS).

For isolated *Chlamydomonas* basal bodies (*Klena et al., 2018*), U-ExM was performed as previously described (*Gambarotto et al., 2019*). Briefly, coverslips were incubated in 1% AA + 0.7% FA diluted in 1X PBS (1X AA/FA) for 5 hr at 37°C prior to gelation in MS supplemented with TEMED and APS (final concentration of 0.5%) for 1 hr at 37°C and denaturation for 30 min at 95°C. Specifically, gels were stained for 3 hr at 37°C with primary antibodies against tubulin monobody AA345 (1:500, scFv-F2C, Alpha-tubulin) (*Nizak et al., 2003*) and POC16 (1:100) (*Hamel et al., 2017*) or POB15 (1:100) (*Hamel et al., 2017*) diluted in 2% PBS/BSA. Gels were washed 3 × 10 min in PBS with 0.1% Tween 20 (PBST) prior to secondary antibodies incubation for 3 hr at 37°C and 3 × 10 min washes in PBST. Gels were expanded in 3 × 150 mL ddH20 before imaging.

Human U2OS cells were grown on 12 mm coverslips and processed as previously described (*Le Guennec et al., 2020*). Briefly, coverslips were incubated for 5 hr in 2% AA + 1.4% FA diluted in 1X PBS (2X AA/FA) at 37°C prior to gelation in MS supplemented with TEMED and APS (final concentration of 0.5%) for 1 hr at 37°C. Denaturation was performed for 1h30 at 95°C and gels were stained as described above. The following primary antibodies were used: tubulin monobodies AA344 (1:250, scFv-S11B, Beta-tubulin) and AA345 (1:250, scFv-F2C, Alpha-tubulin) (*Nizak et al., 2003*), rabbit polyclonal anti-POC1B (1:250, PA5-24495, ThermoFisher), rabbit polyclonal anti-POC5 (1:250, A303-341A, Bethyl), rabbit polyclonal anti-FAM161A (1:250) (*Le Guennec et al., 2020*), mouse monoclonal anti-Centrin (1:250, clone 20H5, 04–1624, Merck Millipore), rabbit polyclonal anti-CEP135 (1:250, 24428–1-AP, Proteintech), rabbit polyclonal anti-PolyE (1:500, AG-25B-0030, AdipoGen), rabbit polyclonal anti-FGFR1OP (FOP1) (1:250, HPA071876, Sigma Life Science), rabbit polyclonal anti-CEP164 (1:250, 22227–1-AP, Proteintech) rabbit polyclonal anti-WDR90 (1:100, NovusBio NBP2-31888). Specifically, as WDR90 staining is weak and dotty, incubation with anti-WDR90 antibodies was performed overnight at 37°C.

Note that for the protein mapping in *Figure 1*, the localisation of the proteins is relative to the epitopes detected by the antibodies used in this approach.

The following secondary antibodies were used: goat anti-rabbit Alexa Fluor 488 IgG H+L (1:400, A11008) and goat anti-mouse Alexa Fluor 568 IgG H+L (1:250, A11004) (Invitrogen, ThermoFisher).

For each gel, a caliper was used to accurately measure its expanded size ($Ex_{size}$ in mm). The gel expansion factor (X factor) was obtained by dividing $Ex_{size}$ by 12 mm, which corresponds to the size of the coverslips use for sample seeding. Thus, X factor = $Ex_{size}$ (mm)/12(mm). The table below shows the $Ex_{size}$ and X factor for all the gels used in this study.

| Gel | siControl<br>$Ex_{size}$ (X factor) | siWDR90<br>$Ex_{size}$ (X factor) |
|---|---|---|
| POC1B (n = 1) | 53 mm (4.42) | 52 mm (4.33) |
| POC1B (n = 2) | 49 mm (4.08) | 50.5 mm (4.21) |
| POC1B (n = 3) | 50.5 mm (4.21) | 50.5 mm (4.21) |
| FAM161A (n = 1) | 50 mm (4.16) | 50 mm (4.16) |
| FAM161A (n = 2) | 50 mm (4.16) | 51 mm (4.25) |
| FAM161A (n = 3) | 50 mm (4.16) | 50 mm (4.16) |
| POC5 (n = 1) | 51 mm (4.25) | 50.5 mm (4.21) |
| POC5 (n = 2) | 50 mm (4.16) | 50 mm (4.16) |
| POC5 (n = 3) | 50.5 mm (4.21) | 49 mm (4.08) |
| Centrin (n = 1) | 50 mm (4.16) | 50 mm (4.16) |
| Centrin (n = 2) | 50 mm (4.16) | 50 mm (4.16) |
| Centrin (n = 3) | 49 mm (4.08) | 49 mm (4.08) |

Pieces of gels were mounted on 24 mm round precision coverslips (1.5H, 0117640, Marienfeld) coated with poly-D-lysine for imaging. Image acquisition was performed on an inverted Leica TCS SP8 microscope or on a Leica Thunder DMi8 microscope using a 63 × 1.4 NA oil objective with Lightening or Thunder SVCC (small volume computational clearing) mode at max resolution, adaptive as 'Strategy' and water as 'Mounting medium' to generate deconvolved images. 3D stacks were acquired with 0.12 µm z-intervals and an x, y pixel size of 35 nm.

## Image analysis

For centrioles counting, immunofluorescences were analyzed on a Leica epifluorescence microscope or on a Leica Thunder DMi8 microscope.

For fluorescence intensity, maximal projections were used using Fiji (*Schindelin et al., 2012*).

Confocal centrosomal intensities were assessed using an area of 20 pixels on Fiji. For each experiment, control values were averaged and all individual measures for control and treated conditions were normalized accordingly to obtain the relative intensity (A.U.). Normalized individual values were plotted on GraphPadPrism7.

Confocal centriolar intensities were assessed by individual plot profil (25 points) on each pair of mature centrioles. For each experiment, the average (Av) of control values was calculated and all individual measures for control and treated conditions were normalized on Av to obtain the relative intensity (A.U.). An average of all normalized measures was generated and plotted in GraphPadPrism7.

For U-ExM data, length coverage quantification was performed as previously published in *Le Guennec et al., 2020*.

For top views, a measurement from the exterior to the interior of the centriole was performed on each microtubule triplet displaying a resolved signal for both tubulin and the core protein. For each tubulin measurement, the position (x-value) of the maximal fluorescence intensity of the core protein was aligned individually to the position of the respective tubulin maximal intensity. All individual values of distance were plotted and analyzed in GraphPadPrism7.

Measurements of diameter in siControl and siWDR90 conditions were performed on S-phase mature centrioles imaged in lateral view. Briefly, lines of 50 pixels thickness were drawn within the proximal, central and distal regions defined in respect with the position of inner core proteins POC5 and FAM161A. Proximal region was then defined as the portion of the centriole below staining of POC5 or FAM161A and the distal region as above. In the siWDR90 condition, proximal region was defined as below the remaining belt of POC5 of FAM161A, the core region was measured just

above the remaining belt and the distal region as the last 100 nm of the centriole. The Fiji plot profile tool was used to obtain the fluorescence intensity profile from proximal to distal for tubulin and the core protein from the same line scan.

Roundness was calculated on perfectly imaged top views of centrioles by connecting tubulin peaks on ImageJ.

## Statistical analysis

No statistical method was used to estimate sample size. The comparison of two groups was performed using a two-sided Student's t-test or its non-parametric correspondent, the Mann-Whitney test, if normality was not granted because rejected by Pearson test The comparisons of more than two groups were made using one- or two-way ANOVAs followed by post-hoc tests (Holm Sidak's multiple comparisons) to identify all the significant group differences. N indicates independent biological replicates from distinct samples. Every experiment was performed at least three times independently on different biological samples unless specified. Data are all represented as scatter or aligned dot plot with centerline as mean, except for percentages quantifications, which are represented as histogram bars. The graphs with error bars indicate SD (+/-) and the significance level is denoted as usual (*p<0.05, **p<0.01, ***p<0.001, ****p<0.0001). All the statistical analyses were performed using Excel or Prism7 (Graphpad version 7.0a, April 2, 2016).

## Acknowledgements

We thank Michel Bornens, Eloise Bertiaux and Nikolai Klena for critical reading of the manuscript, Kanh Huy Bui for sharing his cryo-EM map EMD-20858 and Juliette Azimzadeh for sharing the construct GFP-WDR90-FL. We thank Susanne Dutcher who performed the genetic crosses on the *Chlamydomonas reinhardtii* strains *poc16m504* and *m555*. We thank the BioImaging Center and PFMU at Unige. This work is supported by the Swiss National Science Foundation (SNSF) PP00P3_157517 and PP00P3_187198 (to PG) and 31003A_166608 (to MOS), and by the European research Council ERC ACCENT StG 715289 attributed to Paul Guichard as well as the PhD Booster of the University of Geneva attributed to ES. Further information and requests for resources and reagents should be directed to and will be fulfilled by Virginie Hamel (virginie.hamel@unige.ch) and Paul Guichard (paul.guichard@unige.ch).

## Additional information

### Funding

| Funder | Grant reference number | Author |
|---|---|---|
| Swiss National Science Foundation | PP00P3_157517 | Paul Guichard |
| Swiss National Science Foundation | 31003A_166608 | Michel O Steinmetz |
| European Research Council | StG 715289 | Paul Guichard |
| Swiss National Science Foundation | PP00P3_187198 | Paul Guichard |

The funders had no role in study design, data collection and interpretation, or the decision to submit the work for publication.

### Author contributions

Emmanuelle Steib, Marine H Laporte, Data curation, Formal analysis, Investigation, Visualization, Writing - review and editing; Davide Gambarotto, Data curation, Formal analysis, Visualization; Natacha Olieric, Celine Zheng, Susanne Borgers, Vincent Olieric, France Koll, Anne-Marie Tassin, Data curation, Formal analysis; Maeva Le Guennec, Formal analysis, Visualization; Michel O Steinmetz, Formal analysis, Supervision; Paul Guichard, Conceptualization, Formal analysis, Supervision, Funding acquisition, Visualization, Writing - original draft, Writing - review and editing; Virginie Hamel,

Conceptualization, Data curation, Formal analysis, Supervision, Investigation, Writing - original draft, Project administration, Writing - review and editing

## Author ORCIDs

Emmanuelle Steib (ID) https://orcid.org/0000-0002-5897-1964
Marine H Laporte (ID) https://orcid.org/0000-0002-7856-6763
Celine Zheng (ID) https://orcid.org/0000-0002-8044-9371
Paul Guichard (ID) https://orcid.org/0000-0002-0363-1049
Virginie Hamel (ID) https://orcid.org/0000-0001-5092-2343

## Decision letter and Author response

Decision letter https://doi.org/10.7554/eLife.57205.sa1
Author response https://doi.org/10.7554/eLife.57205.sa2

## Additional files

### Supplementary files

• Transparent reporting form

### Data availability

All data generated or analysed during this study are included in the manuscript and supporting files.

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

# Appendix 1

## Supplemental methods

### Protein alignment

The protein sequences were aligned using Clustal Omega and the secondary structure elements were predicted using Phyre 2, PONDR and XtalPred-RF.

### 3D model

The *Chlamydomonas* POC16 model was prepared using *Phyre2* (Kelley 2015 Nature Protocols) and refined against the FAP20 cryo-EM map EMD_20858 using *phenix.real_space_refine* (Afonine 2018 ActaD). Superposition of the POC16 model excluding flexible loops against FAP20 was done using *COOT* (Emsley 2010 ActaD) and yielded a Root Mean Square Deviation (RMSD) value of 1.6 Angs. The figures were prepared using *ChimeraX* (Goddard 2018 Protein Science).

### PtPOC16 antibody purification

To generate anti-PtPOC16 antibody, a fragment encoding amino acids 2–210 was used for rabbit immunization (Eurogentec). Antibodies were subsequently affinity-purified over a column of PtPOC16(2-210) immobilized on Affi-Gel 10 (Bio-Rad Laboratories) and dialyzed against PBS/5% glycerol.

### Immunofluorescence in *Paramecium tetraurelia*

Immunofluorescence was performed according to *Beisson et al., 2010*. Briefly, Paramecia were permeabilized for 5 min in 0.5% saponin in PHEM Buffer (PIPES 60 mM, HEPES 25 mM, EGTA 10 mM, 2 mM MgCl2 pH 6.9) and fixed in 2% paraformaldehyde (PFA) for 10 min. Cells were washed 3 × 10 min in PHEM-saponin buffer and stained with primary antibodies against POC16 (1:50) and tubulin 1D5 (1:10) for 30 min at room temperature. Cells were incubated with secondary antibodies for 20 min, washed twice in PHEM-saponin prior to a last wash in TBST-BSA supplemented with Hoechst 2 mg/mL.

Imaging was performed on a Zeiss LSM700 confocal microscope with a PlanApo 40x oil immersion objective (NA 1.4) and optical sections were acquired every 0.33 μm, then projected together using ImageJ.

### In vitro POC16 microtubule decoration

In vitro stabilized Taxol-microtubules were prepared in MES-BRB80 derived buffer in contrast to K-PIPES-BRB80 to allow POC16(1-295) protein solubility. Samples were then processed similarly to WDR90-N(1-225).

### Human cells cold shock treatment

U2OS cells grown on 15 mm coverslips and transiently overexpressing mCherry-WDR90-N(1-225)RR for 24 hr were placed in 4°C PBS for an hour on ice and fixed in −20°C methanol. Coverslips were processed for immunofluorescence using primary antibodies against mCherry (1:500) and anti-tubulin DM1α (1:1000).

### Mitotic shake off

RPE1 p53- cells were seeded in T300 flasks the day before shake off. Flasks were shaken vigorously to detach mitotic cells collected in medium. Cells were pelleted by centrifugation for 5 min at 1000 rpm and suspended in 10 nM EdU containing medium prior to seeding in six well plates onto 15 mm coverslips. Cells were fixed at different time points and processed in parallel for immunofluorescence or FACS analysis.

## FACS analysis

Cells were processed similarly to Macheret et al 2018. Post-mitotic cells were washed 2x with PBS then permeabilized and treated with Click-EdU-Alexa 647 (Carl Roth EdU Click FC-647, ref 7783.1) according to manufacturer's instruction. Genomic DNA was stained with propidium iodide (Sigma, Cat. No. 81845) in combination with RNase (Roche, Cat. No. 11119915001). EdU-DNA content profiles were acquired by flow cytometry (Gallios, Beckman Coulter) to assess the percentage of cells that entered S phase in each condition at each time point.

## PCM1 depletion using siRNA

Stable inducible GFP-WDR90 U2OS cells were plated in doxycycline containing–medium onto 15 mm coverslips in a six well plate and 20 nM silencer select pre-designed siRNA ADCSU9L was transfected using Lipofectamine RNAimax (Thermo Fischer Scientific). Medium was changed 4 hr post-transfection and cells were analyzed 48 hr post-transfection.

