## [Decision Letter]

**Acceptance summary:**

This study identifies POC16/WDR90 as an evolutionarily conserved component that interacts with centriolar wall microtubules and is crucial for the assembly of proteins that are part of the inner scaffold structure. In the absence of WDR90, centriole wall defects are observed, suggesting that WDR90 is crucial for centrioles to resist mechanical stress that may occur during spindle assembly and cell division or during ciliary beating.

**Decision letter after peer review:**

Thank you for submitting your article "WDR90 is a centriolar microtubule wall protein important for centriole architecture integrity" for consideration by *eLife*. Your article has been reviewed by three peer reviewers, including Jens Lüders as the Reviewing Editor and Reviewer #1, and the evaluation has been overseen by a Reviewing Editor and Piali Sengupta as the Senior Editor. The following individuals involved in review of your submission have agreed to reveal their identity: Meng-Fu Bryan Tsou (Reviewer #3).

The reviewers have discussed the reviews with one another and the Reviewing Editor has drafted this decision to help you prepare a revised submission.

Summary:

In their manuscript Steib et al., characterize the conserved centriole protein WDR90/POC16. Previously, the authors demonstrated that WDR90 localizes to the central inner centriole wall in human/*Chlamydomonas* cells and that its depletion impairs cilia/flagella formation. Here, they demonstrate that WDR90 may directly bind to the centriole wall, where it could link the A- and B-tubules, separating itself from other components of the inner scaffold (POC1B, FAM161A, POC5 and centrin). Using cellular localization, in vitro binding assays, and EM analysis, the authors further show that WDR90 can directly associate with pure MT polymers and free tubulin dimers, suggesting a role of POC16/WDR90 in linking the MT wall to the inner scaffold of the centriole in the central core region. Consistently, functional studies showed that in the absence of WDR90, several inner scaffold components fail to localize to centrioles, and that centrioles depleted of WDR90 become unstable, often losing structural integrity.

Overall, this is a nicely presented follow-up of the group's previous work that reveals new detail about the organization of proteins at the inner centriole wall, which is important for our understanding of centriole structural integrity. In particular the localization and biochemical studies are solid and of high quality. However, there is also some overlap with previously published work regarding the mapping of inner scaffold components (Le Guennec et al., 2020). Moreover, the relationships between WDR90, POC5 and other inner wall proteins in human cells and differences in RNAi phenotypes could be elucidated in more detail.

Essential revisions:

1) Previous work suggests FOP1, glutamylaton, BLD10, and POC1 play roles in stability. Do these proteins (besides POC1) play a similar or different role, or are affected by reduction of WDR90?

2) Loss of POC5 in human cells results in short centrioles that fail to elongate (Azimzadeh et al., 2009), whereas WDR90-depleted centrioles, in which POC5 is also lost, become over-elongated (this study). It would be very informative if the authors can address this discrepancy. For example, where is WDR90 in POC5-depleted centrioles? What does the centriole look like (longer or shorter) when both WDR90 and POC5 are gone? Does WDR90 really function upstream of POC5 (or other inner scaffold components) as the model suggests?

3) Does the distorted shape of centrioles in WDR90-depleted cells, shown in cross sections in Figure 6, truly reflect the shape of WDR90-depleted centrioles prior to expansion or is the distortion a result of the expansion technique, performed on structurally compromised centrioles? Is this also seen by EM analysis?

The authors should classify the phenotypes and show the frequency for each category of abnormal centrioles by U-ExM and provide EM analyses if available.

---

## [Author Response]

Summary:In their manuscript Steib et al., characterize the conserved centriole protein WDR90/POC16. Previously, the authors demonstrated that WDR90 localizes to the central inner centriole wall in human/Chlamydomonas cells and that its depletion impairs cilia/flagella formation. Here, they demonstrate that WDR90 may directly bind to the centriole wall, where it could link the A- and B-tubules, separating itself from other components of the inner scaffold (POC1B, FAM161A, POC5 and centrin). Using cellular localization, in vitro binding assays, and EM analysis, the authors further show that WDR90 can directly associate with pure MT polymers and free tubulin dimers, suggesting a role of POC16/WDR90 in linking the MT wall to the inner scaffold of the centriole in the central core region. Consistently, functional studies showed that in the absence of WDR90, several inner scaffold components fail to localize to centrioles, and that centrioles depleted of WDR90 become unstable, often losing structural integrity.Overall, this is a nicely presented follow-up of the group's previous work that reveals new detail about the organization of proteins at the inner centriole wall, which is important for our understanding of centriole structural integrity. In particular the localization and biochemical studies are solid and of high quality. However, there is also some overlap with previously published work regarding the mapping of inner scaffold components (Le Guennec et al., 2020). Moreover, the relationships between WDR90, POC5 and other inner wall proteins in human cells and differences in RNAi phenotypes could be elucidated in more detail.

We would like to thank the editor for efficient editorial processing and appreciate the reviewer’s comments and suggestions to further improve the quality of our manuscript. We have made all efforts to address them and detail our replies below.

Essential revisions:1) Previous work suggests FOP1, glutamylaton, BLD10, and POC1 play roles in stability. Do these proteins (besides POC1) play a similar or different role, or are affected by reduction of WDR90?

To address this interesting point, we conducted additional experiments and assess the localization of FOP1, CEP135 (BLD10) as well as glutamylation using PolyE antibodies in siControl and siWDR90 treated expanded U2OS cells. Interestingly, we found that Polyglutamylation decorates the centriolar microtubule wall as previously reported (Mahecic et al., 2020 and Sullenberger et al., 2020) while CEP135 displayed a more complex localization spanning the entire centriole with a clear proximal enrichment. Finally, FOP1 decorated the external central/distal region of the centriole, forming a surrounding shell around the centriole. Overall, we found that glutamylation and CEP135 localization were not affected by the depletion of WDR90. These results suggest that the stability defects observed upon WDR90 depletion are not linked to the protein CEP135 nor to glutamylation defects. In addition, we observed a slight impact on FOP1 distribution that appears more dispersed in WDR90-depleted condition. The new acquired data are now included in the revised manuscript as a new Figure 4—figure supplement 1 as well as in the revised text (subsection “WDR90 depletion leads to a loss of inner scaffold components and to centriole fracture”).

2) Loss of POC5 in human cells results in short centrioles that fail to elongate (Azimzadeh et al., 2009), whereas WDR90-depleted centrioles, in which POC5 is also lost, become over-elongated (this study). It would be very informative if the authors can address this discrepancy.

Again, we would like to thank the reviewers for pointing out this interesting comment. To address it, we depleted POC5 using the previously described *poc5* siRNA (Azimzadeh et al., 2005) and analyzed the resulting phenotype by expansion microscopy. We first demonstrate that POC5 is efficiently depleted upon siRNA treatment both by standard immunofluorescence microscopy as well as by U-ExM (Figure 6 and Figure 6—figure supplement 1). We next analyzed POC5-depleted mature centrioles by U-ExM and found, similarly to what we report for WDR90 depletion, that these depleted centrioles were longer (Figure 6). This important finding suggests greatly that depletion of the inner scaffold proteins, such as POC5 or WDR90, can lead to a slight centriole elongation. Importantly, we verified that in mitotic cells, similarly to the published work of Azimzadeh et al., procentrioles were shorter than control cells, validating the POC5 depletion condition (Figure 6—figure supplement 1). We also analyzed centriole post-mitosis and found that these procentrioles converted into daughter centrioles are then longer. We do think that the centriole over-elongation happens during mitosis, explaining the discrepancy between our results and the previous one published by the Bornens Lab. This point is now reported in subsection “WDR90/POC5 co-depletion enhances centriole architecture abnormalities” of the revised manuscript.

For example, where is WDR90 in POC5-depleted centrioles?

To answer this question, we analyzed the distribution of WDR90 upon POC5 depletion both by standard immunofluorescence microscopy and expansion microscopy (Figure 6 and Figure 6—figure supplement 1). Importantly, we found that WDR90 is not affected by POC5 depletion, suggesting that WDR90 is upstream of POC5 (subsection “WDR90/POC5 co-depletion enhances centriole architecture abnormalities”).

What does the centriole look like (longer or shorter) when both WDR90 and POC5 are gone? Does WDR90 really function upstream of POC5 (or other inner scaffold components) as the model suggests?

This is a great question! We co-depleted WDR90 and POC5 and assess the size of centrioles under these conditions using U-ExM. Remarkably, we found that under this condition, the remaining belt of POC5 observed upon WDR90 depletion is now completely gone and we also find a significant proportion of mother and daughter centrioles depleted. Remarkably, we found that the phenotype of the co-depletion led to overly long centrioles (Figure 6 and Figure 6—figure supplement 1). More importantly, the number of broken centrioles are now about 30% upon double depletion while we found only 10% of depleted centrioles with single depletion (WDR90 or POC5) (Figure 5 and Figure 6—figure supplement 2). These results confirm that the removal of inner scaffold components such as WDR90 and POC5 affects centriole integrity.

The results from these experiments are presented in Figure 5, Figure 6, Figure 6—figure supplement 1 and Figure 6—figure supplement 2 and described in the revised manuscript (subsection “WDR90/POC5 co-depletion enhances centriole architecture abnormalities”).

3) Does the distorted shape of centrioles in WDR90-depleted cells, shown in cross sections in Figure 6, truly reflect the shape of WDR90-depleted centrioles prior to expansion or is the distortion a result of the expansion technique, performed on structurally compromised centrioles? Is this also seen by EM analysis?The authors should classify the phenotypes and show the frequency for each category of abnormal centrioles by U-ExM and provide EM analyses if available.

We understand the reviewer’s concerns and provide now the classification of the phenotypes, showing the frequency of each category in the Figure 5—figure supplement 1. We have notably quantified the percentage of daughter centrioles that appears in top view and found it to be minority with only 5.3% of the cases observed (Figure 5—figure supplement 1A). Consistently, in WDR90-depleted cells, 4% (17/356 counted centrioles) of depleted centrioles were oriented in top views (Figure 5—figure supplement 1B). Importantly, 95% of these were abnormal, with an affected roundness, indicating that most of the depleted daughter centriole in top view had lost the canonical roundness shape of centrioles (Figure 5—figure supplement 1C).

Although we do not believe that this results from a distortion owing to the U-ExM technique, we cannot exclude formally that the more fragile/unstable centrioles in WDR90-depleted condition are not affected by the technique. We now stated this possibility in the discussion of the revised manuscript (Discussion section).

Lastly, concerning the EM analysis, we do not provide such examples owing to the technical difficulties in finding perfectly well-oriented top-view centrioles that are depleted of WDR90 as explained above.